# HAWKBENCH: Investigating Resilience of RAG Methods on Stratified Information-Seeking Tasks

**Hongjin Qian**[1,2], **Zheng Liu**[2,3],[*] **Chao Gao**[6], **Yankai Wang**[5]
**Defu Lian**[5], **Zhicheng Dou**[4]
[1] Peking University
[2] Beijing Academy of Artificial Intelligence
[3] Hong Kong Polytechnic University
[4] Renmin University of China
[5] University of Science and Technology of China
[6] Hong Kong University of Science and Technology
`{chienqhj,zhengliu1026}@gmail.com`

## Abstract

In real-world information-seeking scenarios, users have dynamic and diverse needs, requiring RAG systems to demonstrate adaptable resilience. To comprehensively evaluate the resilience of current RAG methods, we introduce HawkBench, a human-labeled, multi-domain benchmark designed to rigorously assess RAG performance across categorized task types. By stratifying tasks based on information-seeking behaviors, HawkBench provides a systematic evaluation of how well RAG systems adapt to diverse user needs. Unlike existing benchmarks, which focus primarily on specific task types (mostly factoid queries) and rely on varying knowledge bases, HawkBench offers: (1) systematic task stratification to cover a broad range of query types, including both factoid and rationale queries, (2) integration of multi-domain corpora across all task types to mitigate corpus bias, and (3) rigorous annotation for high-quality evaluation. HawkBench includes 1,600 high-quality test samples, evenly distributed across domains and task types. Using this benchmark, we evaluate representative RAG methods, analyzing their performance in terms of answer quality and response latency. Our findings highlight the need for dynamic task strategies that integrate decision-making, query interpretation, and global knowledge understanding to improve RAG generalizability. We believe HawkBench serves as a pivotal benchmark for advancing the resilience of RAG methods and their ability to achieve general-purpose information seeking. We host our codes and data in *this repository*.

## 1 Introduction

Large Language Models (LLMs) excel in general reasoning and knowledge-based tasks but often struggle with timeliness and knowledge coverage gaps, particularly in specialized domains and user-specific data [OpenAI, 2023, DeepSeek-AI, 2024]. To address these limitations, incorporating external knowledge has become a common approach, with Retrieval-Augmented Generation (RAG) emerging as an effective solution to enhance factual accuracy and adaptability [Zhu et al., 2024].

During the information-seeking process using RAG, users may have a wide range of information needs, from simple factoid retrieval to more complex rationale-based queries [Qian et al., 2025, Zhao et al., 2024a]. This versatility requires RAG systems to possess diverse capabilities, including accurate referencing and advanced reasoning skills.

---

[*]Corresponding author.

39th Conference on Neural Information Processing Systems (NeurIPS 2025) Track on Datasets and Benchmarks.

Recent advancements in RAG methods have enhanced vanilla RAG systems by targeting specific advanced capabilities. For instance, some methods focus on improving multi-hop reasoning to handle tasks with implicit information intents [Zhao et al., 2024b, Xu et al., 2024], while others address information aggregation tasks by constructing intermediate structures, such as graphs or memory modules, to better integrate relevant information [Qian et al., 2025, Edge et al., 2024].

While these advancements enable RAG systems to effectively leverage external knowledge for specific tasks, their ability to generalize across diverse scenarios remains uncertain. A recent survey categorizes external knowledge-based tasks into distinct levels, emphasizing that no single method can effectively address all query types [Zhao et al., 2024a]. This suggests that current RAG methods lack the resilience required for general-purpose information-seeking tasks, highlighting the need for a systematic evaluation of RAG methods across a broad range of information-seeking tasks, examining the resilience of these methods when faced with information-seeking tasks in any form.

Existing public benchmarks for RAG evaluation focus narrowly on isolated dimensions of information-seeking tasks. For instance, LegalBench-RAG evaluates information-seeking tasks in the legal domain [Guha et al., 2023], MutiHop-RAG tests multi-hop reasoning [Tang and Yang, 2024], and CRAG emphasizes comprehensive evaluation on factual QA tasks [Yang et al., 2024]. While these benchmarks excel in their targeted domains, they collectively fail to assess the resilience of RAG methods across stratified task types due to three critical limitations:

**First, fragmented evaluation protocols**. Current benchmarks are siloed by design, each prioritizing distinct query types. This specialization creates inconsistent evaluation criteria, hindering fair comparisons of RAG performance across diverse task categories. **Second, domain bias and knowledge leakage**. Many benchmarks rely on heterogeneous knowledge bases (e.g., Wikipedia and web snippets), leading to corpus-dependent performance gaps that obscure true method capabilities. Worse, LLMs are often pretrained on these same sources (e.g., Wikipedia), inflating benchmark scores through memorization rather than genuine retrieval-augmented reasoning. **Third, limited query diversity.** Most benchmarks disproportionately emphasize factoid questions (e.g., "When was Einstein born?"), neglecting rationale-based queries (e.g., "Explain how relativity revolutionized physics") that require synthesis and contextual analysis. This narrow focus misaligns with real-world user needs, where information-seeking behaviors span both factual lookup and complex reasoning.

HawkBench is characterized by the following key features:

**Domain Thoroughness** – We curate raw texts from a diverse range of sources—including professional textbooks, academic papers, financial reports, legal contracts, and novels—to ensure that the benchmark reflects real-world information needs. This broad selection captures both general and specialized knowledge, offering a robust foundation for evaluation.

**Systematic Task Stratification** – We systematically define four query types: (1) explicit factoid queries, (2) implicit factoid queries, (3) explicit rationale queries, and (4) implicit rationale queries. This stratification, inspired by Zhao et al. [2024a] with refined modifications, ensures comprehensive task coverage. Importantly, all query types share the same underlying knowledge distribution, allowing for direct and fair performance comparisons across different tasks.

**Rigorous Annotation Quality** – HawkBench employs a hybrid annotation process that leverages both advanced LLMs—specifically GPT-4 and DeepSeek-V3—and human oversight. Initially, LLMs generate query-answer pairs from the curated texts. Expert annotators then evaluate these pairs against predefined stratification levels, refine the answers by correcting inaccuracies, and enhance clarity. This process results in a high-quality dataset of 1,600 annotated test samples, evenly distributed across all task types.

We further validate HawkBench by applying representative RAG methods and performing a comprehensive analysis of their performance in terms of both answer quality and response latency. Our empirical results reveal that while current RAG methods excel in specific tasks, they generally lack overall resilience. Enhancing their adaptability will require dynamic task strategies that integrate decision-making, query interpretation, and a holistic understanding of global knowledge.

Our contributions are as follows: (1) We introduce HawkBench, a high-quality benchmark with stratified tasks designed to assess the resilience of RAG methods for general-purpose information-seeking. (2) We conduct a comprehensive empirical evaluation of recent RAG methods on HawkBench,

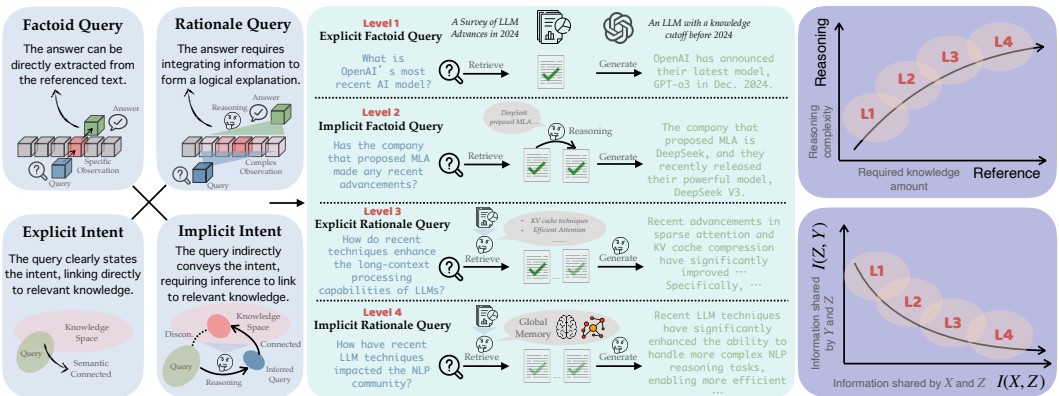

Figure 1: Query Stratification of HAWKBENCH. To account for referencing difficulty, we categorize tasks into queries with explicit intent and implicit intent. Regarding reasoning, tasks are categorized into factoid queries and rationale queries. By combining these two categorizations, we stratify information-seeking tasks into four levels.

enabling a side-by-side comparison of their capabilities. (3) We propose insights and strategies to improve the generalizability and adaptability of current RAG methods.

## 2 HawkBench

### 2.1 Preliminary

Recent advancements in large language models (LLMs) have popularized the Retrieval-Augmented Generation (RAG) approach, which leverages external knowledge to perform specific tasks. In RAG, a generation model $\theta(\cdot)$ and a retrieval model $\gamma(\cdot)$ collaborate to produce a final response $\mathcal{Y}$. Formally, the process is expressed as:

$$\mathcal{Y} = \theta(q, \mathcal{Z}), \quad \mathcal{Z} = \gamma(q, \mathcal{X}), \tag{1}$$

where $q$ denotes the input query, $\mathcal{X}$ represents the external knowledge base, $\mathcal{Z}$ is the retrieved relevant information, and $\mathcal{Y}$ is the generated answer.

This RAG framework can be viewed as an information-refinement process following the *Markov chain*: $\mathcal{X} \rightarrow \mathcal{Z} \rightarrow \mathcal{Y}$. As information passes through each stage, it is progressively distilled, leading to the inequality $I(\mathcal{X}, \mathcal{Z}) \geq I(\mathcal{Y}, \mathcal{Z})$, where $I(\cdot)$ denotes mutual information. Ideally, the retrieval step should extract a $\mathcal{Z}$ that is both *sufficient*—containing all the information necessary to generate $\mathcal{Y}$—and *minimal*—excluding irrelevant details from $\mathcal{X}$. In fact, the condition $I(\mathcal{X}, \mathcal{Z}) = I(\mathcal{Y}, \mathcal{Z})$ would hold if and only if an optimal retrieval output $\mathcal{Z}^*$ exists that perfectly balances these two criteria. Achieving such an optimal $\mathcal{Z}^*$ is challenging due to estimation biases in both the retrieval and generation processes. To better understand these challenges, it is essential to consider two interrelated dimensions:

**Referencing** The retrieval process must determine not only which pieces of information in $\mathcal{X}$ are relevant to the query $q$ but also how much information is required. The *referencing* is straightforward when $q$ explicitly states its intent, as the semantic connections between $q$ and the relevant content in $\mathcal{X}$ are easier to measure. However, for implicit queries—where the intent is not clearly stated—identifying the necessary evidence becomes more complex. Thus, the referencing dimension measures *how to access* the relevant knowledge, capturing both the volume of information needed and its accessibility within the knowledge base.

**Reasoning** Once the retrieval model produces $\mathcal{Z}$, the generation model must process and integrate this information to formulate the final answer $\mathcal{Y}$. For factoid queries, the retrieved information typically aligns closely with the required answer, meaning that the reasoning effort is relatively minimal. In contrast, when the query demands a rationale—requiring the synthesis and integration of multiple pieces of information—the generation process must engage in more complex in-context

reasoning. Therefore, the reasoning dimension measures *how to utilize* the relevant knowledge, reflecting the cognitive effort needed to bridge the gap between the retrieved data and the final, coherent response.

To systematically analyze the difficulty of information-seeking tasks within the RAG framework, we decompose queries along these two dimensions. As shown in Figure 1 (left), we categorize tasks based on: **Referencing:** Whether the query explicitly or implicitly conveys its intent, thereby affecting the ease with which relevant information can be identified. **Reasoning:** Whether the task involves straightforward fact extraction or requires integrating information to form a reasoned response. By combining these dimensions, we define four levels of information-seeking tasks, each posing unique challenges to the RAG pipeline, as outlined in the next section.

## 2.2 Query Stratification

In Figure 1 (middle), we illustrate our query stratification, presenting the four query types below.

**Level 1: Explicit Factoid Query**    Level 1 queries exhibit an explicitly stated information-seeking intent and typically require minimal reasoning. The answer is directly available in the retrieved text. For instance, the query

> *"What is OpenAI's most recent AI model?"*

clearly specifies its intent, allowing the retrieval system to easily locate the pertinent information. The generator can then extract the final answer with little or no additional reasoning.

**Level 2: Implicit Factoid Query**    Level 2 queries present an implicit information-seeking intent, which necessitates an extra step to resolve the reference before the answer can be extracted. Consider the query

> *"Has the company that proposed MLA made any recent advancements?"*

The query does not directly name the company. The system must first infer that "the company that proposed MLA" refers to, for example, DeepSeek. Once this implicit reference is established, the relevant knowledge can be retrieved, and the answer can be extracted with minimal reasoning. Thus, Level 2 queries require additional referencing effort compared to Level 1, while the reasoning for answer extraction remains straightforward.

**Level 3: Explicit Rationale Query**    In Level 3 queries, the intent is explicitly stated, but there exists a semantic gap between the query and the relevant information. Although the query clearly indicates what is being asked, the final answer is not directly extractable from a single text fragment and requires synthesizing information from multiple sources. For example, the query

> *"How do recent techniques enhance the long-context processing capabilities of LLMs?"*

explicitly requests an explanation. However, the necessary rationale is dispersed across several texts. This scenario demands a more complex retrieval process, possibly aided by structured representations (e.g., graphs), and a generator capable of synthesizing the information into a coherent answer.

**Level 4: Implicit Rationale Query**    Level 4 queries pose the highest challenge as they involve both an implicit intent and the need to generate a global explanation. For example, the query

> *"How have recent LLM techniques impacted the NLP community?"*

requires the system to first infer the underlying intent and then integrate diverse pieces of information across the entire knowledge base to form a comprehensive explanation. This task demands extensive referencing to identify loosely connected yet relevant content and significant reasoning to synthesize a unified, high-level response.

## 2.3 Comparison of the Four Query Levels

In Figure 1 (right), we compare the four query levels across two aspects: *Reference* and *Reasoning*. First, in terms of *Reference*, the amount of relevant knowledge required increases from Level 1

to Level 4 queries, reflected in the mutual information between the knowledge base and retrieved knowledge, $I(\mathcal{X}, \mathcal{Z})$. Level 1 queries require minimal knowledge, as answers are directly extractable from a few text chunks. In contrast, higher-level queries, such as Level 3 and Level 4, require synthesizing information from a broader range of texts. Second, in terms of *Reasoning*, complexity increases across levels due to the growing semantic gap between retrieved knowledge and the final answer. For Level 1 queries, reasoning is minimal, but for Level 3 and Level 4 queries, more reasoning is needed to connect multiple, loosely connected pieces of information. This is reflected in the decreasing mutual information $I(\mathcal{Z}, \mathcal{Y})$ as redundant information is filtered out during refinement.

These varying requirements for referencing and reasoning present significant challenges for current RAG systems, which struggle to adapt to the diversity of information-seeking tasks. There is no one-size-fits-all solution, as each task demands distinct capabilities. This underscores the necessity of benchmarking current RAG methods across a broad range of tasks to better assess their resilience.

## 2.4 Construction

**Corpus Collection**   While most current LLMs are proficient in general world knowledge due to their training on large-scale corpora, they often lack coverage in specialized, domain-specific areas. To address this gap, HawkBench incorporates 229 domain-specific texts into its knowledge base. These texts are carefully selected from a larger collection of long texts gathered across diverse domains, which can also serve as a global corpus for retrieval. The selected 229 contexts span a wide range of domains, including professional textbooks (manually labeled into categories such as technology, humanities, art, and science), financial reports, legal contracts, novels, and academic papers. This diverse and comprehensive collection ensures that HawkBench can thoroughly evaluate the domain resilience of RAG methods by covering a broad range of user information needs.

**Annotation Process**   The annotation process for constructing HawkBench follows a systematic approach, as illustrated in Figure 6. The process consists of three key steps:

(1) **Configuration:** The annotator selects the target query level and domain, with assistance from a strong LLM (GPT-4o and DeepSeek-v3).

(2) **Question-Answer Pair Generation:** The system prompts the LLM agent using built-in QA generation prompts to produce initial question-answer pairs. During this step, the system first samples from the knowledge base, selecting a random text span of varying lengths based on the task type. For Level-1 tasks, approximately 1K tokens are used as the context. For Level-2 tasks, we use a retrieval system retrieves the top-10 passages based on the generated L1 query, selecting five passages to prompt the agent to transform explicit factoid queries into implicit intent queries. For Level-3 and Level-4 tasks, up to 120K tokens are sampled as the knowledge context to guide the agent in generating information aggregation queries, with different prompts controlling the process. The codes for annotation system and all built-in prompts are in *this repository*.

(3) **Quality Control:** The annotator reviews the generated question to ensure it aligns with the target task type's definition. If the question is unsuitable, it is discarded. If the question is valid, the annotator evaluates the generated answer for clarity, conciseness, and semantic richness. The answer is then manually edited to ensure high quality.

| L | Discard % | Edit % | Ave. Time | Total Time |
|---|-----------|--------|-----------|------------|
| 1 | 6.7% | 3.5% | 26s | 4.5h |
| 2 | 28.1% | 41.4% | 71s | 23.1h |
| 3 | 25.2% | 47.9% | 183s | 41.5h |
| 4 | 29.1% | 40.6% | 201s | 45.2h |

Figure 2: Statistical Details of Construction.

We employed three PhD students proficient in English as annotators. As shown in Table 2, the difficulty of annotating different task types varies significantly. For Level-1 tasks, most generated QA pairs are valid with only minor edits needed, making this task relatively quick. In contrast, for Levels 2–4, the generated QA pairs are often invalid and discarded, and the quality of the answers generally requires more extensive manual editing. This process results in longer annotation times for higher-level tasks. The total annotation time includes both system latency (primarily due to QA pair generation) and manual annotation work. The three annotators dedicated approximately one week of full-time work to constructing HawkBench, each receiving a salary of around $1,000. Additionally, constructing HawkBench incurred around $597 in GPT-4o usage and $278 in DeepSeek-v3 usage.

**Dataset Distribution**    Table 3 presents the statistical details of HawkBench. The dataset contains 1,600 test samples, derived from 229 context knowledge bases. The compressed file size of Hawk-Bench is only 26MB, making it highly portable for distribution. We have thoroughly reviewed the licenses of all source texts to ensure that they permit redistribution. HawkBench is distributed under the Apache License 2.0.

## 3    Experiment

### 3.1    Baselines and Metrics

To investigate the resilience of RAG methods on HawkBench, we select the following representative baseline methods: **Vanilla RAG:** This method retrieves the top passages as context. **Enhanced RAG Methods:** *HyDE* [Gao et al., 2023] generates a hypothetical document to enhance query retrieval. *RQRAG* [Chan et al., 2024] rewrites the input query into sub-queries to refine retrieval. **Global RAG:** These methods index the knowledge base into an intermediate form to enhance global awareness. This includes memory-based methods such as *MemoRAG* [Qian et al., 2025] and graph-based methods like *GraphRAG* [Edge et al., 2024].

Additionally, we explore the application of long LLMs in HawkBench, including vanilla LLMs, the prompt compression method *Lingua-2* [Pan et al., 2024], and long-context acceleration methods such as *MInference* [Jiang et al., 2024]. All baselines in the main experiments use *Qwen2.5-7B-instruct* as the generator [Team, 2025], with *BGE-M3* as the retriever [Chen et al., 2023] and the top-$k$ set to 5 for all RAG methods.

For Level 1 and Level 2 tasks, which focus on factoid queries, we use *Rouge-L* and lexical F1-score as evaluation metrics. These metrics emphasize surface-form lexical overlap and are well suited for evaluating fact-based answers.

For Level 3 and Level 4 tasks, which involve rationale queries, we introduce a new evaluation metric, denoted as $\mathcal{S}$-F1, to robustly assess sentence-level semantic equivalence between the ground-truth and predicted answers. Specifically, let $A^*$ denote the ground-truth answer and $A$ the predicted answer. We tokenize both $A$ and $A^*$ into sentences $\{s_i\}$ and $\{s_i^*\}$, respectively. Then, $\mathcal{S}$-F1 is defined as:

$$\mathcal{S}\text{-F1}(A, A^*) = \frac{1}{2n} \sum_{i=1}^{n} \mathbb{1}_{\{\text{LLM}(s_i, A^*)=\text{True}\}} + \frac{1}{2m} \sum_{i=1}^{m} \mathbb{1}_{\{\text{LLM}(s_i^*, A)=\text{True}\}}, \qquad (2)$$

where $\mathbb{1}_{\text{condition}}$ is an indicator function that returns 1 if the condition holds and 0 otherwise, $n$ is the number of sentences in $A$, and $m$ is the number of sentences in $A^*$.

Intuitively, $\mathcal{S}$-F1 computes the average of: (1) **Precision:** the proportion of sentences in the predicted answer $s_i \in A$ that are judged by a strong LLM to be semantically supported by the ground-truth answer $A^*$. (2) **Recall:** the proportion of sentences in the ground-truth answer $s_i^* \in A^*$ that are judged to be semantically supported by the predicted answer $A$.

Here, "supported by" means that for a given sentence, the LLM judge determines whether its meaning or rationale is present, possibly rephrased but semantically equivalent, in the other answer. More concretely, we apply the following process: (1) Each predicted answer is split into sentences. For each $s_i \in A$, the LLM judge is prompted to return a binary decision (0/1) indicating whether the content of $s_i$ is covered by $A^*$. (2) Each ground-truth answer is similarly split into sentences, and for each $s_i^* \in A^*$, we query whether its rationale is reflected in $A$.

Compared to lexical F1-score, $\mathcal{S}$-F1 moves beyond surface-form matching and directly evaluates sentence-level semantic alignment between $A$ and $A^*$, making it a more robust metric for rationale-based tasks where lexical overlap alone cannot capture equivalence. For completeness, we also report *Rouge-L* scores alongside $\mathcal{S}$-F1 when evaluating Level 3 and Level 4 tasks.

### 3.2    Main Results

We conduct comprehensive experiments across all baselines, with the full results presented in Table 5. To provide a more detailed analysis, we examine the results from multiple perspectives, offering a deeper understanding of performance across different dimensions.

Table 1: Evaluation performance across four levels, averaged over all domains. The best scores are highlighted in bold, and the second-best scores are underlined.

| Method | Type | LEVEL-1 | | LEVEL-2 | | LEVEL-3 | | LEVEL-4 | |
|---|---|---|---|---|---|---|---|---|---|
| | | Rouge-L | F1 | Rouge-L | F1 | Rouge-L | $\mathcal{S}$-F1 | Rouge-L | $\mathcal{S}$-F1 |
| LLM | Long LLM | 13.0 | 12.9 | 12.9 | 11.5 | 26.2 | 24.0 | 16.9 | 33.2 |
| Lingua-2 | Compression | 11.4 | 11.4 | 12.2 | 11.4 | 23.7 | 23.9 | 15.4 | 25.2 |
| MInference | Accelerating | 11.5 | 11.1 | 12.6 | 11.2 | 25.6 | 24.2 | 17.1 | 33.3 |
| RAG | Standard RAG | 50.9 | 57.5 | 34.0 | 38.6 | 17.9 | 27.3 | 15.3 | 18.3 |
| HyDE | Enhanced RAG | **64.4** | 73.5 | 40.0 | 44.5 | 19.4 | 28.0 | 15.6 | 18.4 |
| RQRAG | Enhanced RAG | 64.2 | **73.6** | **41.1** | **46.8** | 19.7 | 28.6 | 15.4 | 17.4 |
| MemoRAG | Global RAG | 44.8 | 50.2 | 33.7 | 37.3 | **27.3** | **34.1** | 19.0 | **35.0** |
| GraphRAG | Global RAG | 49.3 | 57.4 | 34.0 | 37.0 | 25.3 | 32.5 | **20.6** | 28.7 |

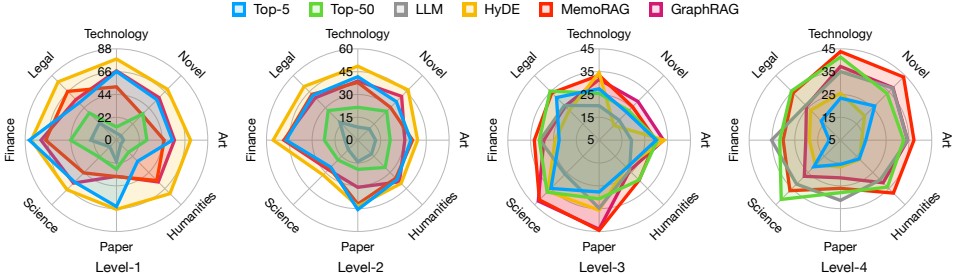

Figure 3: Evaluation performance across four levels and eight domains for selected methods.

**Resilience across Levels** Table 1 presents the performance of all baselines across the four task levels, averaged by domain. From these results, we draw several key insights: (1) Standard RAG and Enhanced RAG methods perform well on factoid queries (Level-1 and Level-2), suggesting that these queries often rely on specific text spans that can be easily located with minimal reasoning or simple enhancements. (2) Global RAG methods underperform on Level-1 and Level-2 tasks but excel on Level-3 and Level-4 tasks. This indicates that global reasoning is not beneficial for factoid queries and may even hinder performance. However, for rationale queries, which require synthesizing information from a broad range of text, global awareness helps gather more comprehensive evidence, leading to improved performance. (3) Directly applying long LLMs to process the entire knowledge base is feasible but underperforms on factoid queries due to over-referencing and redundant noise. However, for rationale queries, long LLMs outperform vanilla RAG methods due to their strong reasoning ability over long contexts. Efficient long-context methods, such as accelerated pre-filling or prompt compression, yield performance comparable to vanilla LLMs.

**Resilience over Domains** Figure 3 presents the experimental results across different levels and domains for selected methods. The results highlight how different methods perform across domain-specific knowledge: (1) For structured knowledge sources, such as financial reports and legal documents, most methods perform well on factoid queries. The inherent clarity and precision of these texts reduce semantic ambiguity, improving retrieval accuracy. (2) For explanatory texts, such as academic papers that focus on providing rationales, global RAG methods excel. Their global awareness enables them to effectively organize and integrate explicit reasoning from the knowledge base. (3) For unstructured knowledge in domains like literature, art, and humanities—where texts contain higher semantic ambiguity—global RAG methods perform better on Level-4 tasks. This suggests that aggregating high-level implicit information is more effective for narrative-based content than for structured knowledge domains.

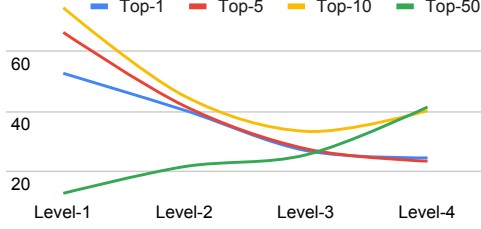

Figure 4: Evaluation performance across four levels for RAG with varying Top-$k$ selections.

**Impact of Top-k** Figure 4 systematically investigates the impact of Top-k selection using vanilla RAG. The results show that while increasing Top-k introduces more knowledge into the generation

process, it also increases redundancy. The trade-off between knowledge recall and precision varies across query levels. Factoid queries rely on precise evidence, and excessive redundancy significantly degrades performance. In contrast, rationale queries benefit from higher recall, as effective information aggregation requires a more comprehensive set of evidence from the knowledge base.

**Efficiency Analysis**    Table 2 presents a comparison of task latency across methods and task levels. The following insights can be drawn from the results:

(1) Standard RAG methods are highly efficient, as the retrieval process is not sensitive to the size of the knowledge base. In contrast, long LLMs and global RAG methods experience a notable increase

in latency across all tasks, while only improving performance on rationale tasks. (2) Long LLMs incur the highest latency for all task types but fail to deliver a clear performance advantage. This suggests that directly using the full knowledge base may not be a proper approach. (3) The graph construction process for GraphRAG relies heavily on robust model APIs, leading to substantial construction latency. However, once the graph is constructed, performance becomes efficient. This indicates that optimizing the process of perceiving the global knowledge base—such as accelerating the graph construction in GraphRAG or memory formation in MemoRAG—could be beneficial for improving performance on rationale queries.

| Level | RAG | HyDE | LLM | MemoRAG | GraphRAG |
|---|---|---|---|---|---|
| 1 | 0.6 | 1.0 | 29.1 | 20.9 | 1.7 $(+\infty)$ |
| 2 | 0.7 | 2.0 | 32.7 | 21.5 | 2.0 $(+\infty)$ |
| 3 | 1.6 | 2.1 | 48.3 | 33.4 | 3.0 $(+\infty)$ |
| 4 | 1.7 | 2.2 | 52.1 | 35.9 | 3.5 $(+\infty)$ |

Table 2: Task latency (queries per second) comparison across methods and levels. Experiments were conducted on an Nvidia A800-80G GPU using the ART dataset. GraphRAG employs GPT-4o for graph construction, which can take up to half an hour, denoted by $+\infty$.

**Retrieval Strategy Analysis**    In addition to comparing different RAG architectures, we further investigate the impact of retrieval strategies on performance across various task levels. Specifically, we evaluate three types of retrievers: **dense retrieval**, **sparse retrieval**, and a **hybrid** approach that combines both. The goal is to understand how the choice of retriever influences the resilience and adaptability of RAG systems under different information-seeking challenges.

Figure 5 presents the performance of representative RAG methods (vanilla RAG, RQRAG, and MemoRAG) using each retrieval strategy across the four task levels in HawkBench. The results demonstrate that retrieval strategy has a substantial impact on downstream performance. Dense and hybrid retrievers consistently outperform sparse retrievers, particularly on rationale-intensive tasks (Levels 3 and 4), where retrieving semantically rich information is crucial. Notably, methods that incorporate additional retrieval cues—such as query rewriting in RQRAG or memory-guided retrieval in MemoRAG—benefit significantly from hybrid retrieval. This suggests that hybrid retrievers enhance the likelihood of capturing diverse, relevant evidence when guided by auxiliary signals. These findings underscore the importance of retrieval design in RAG pipelines, especially when

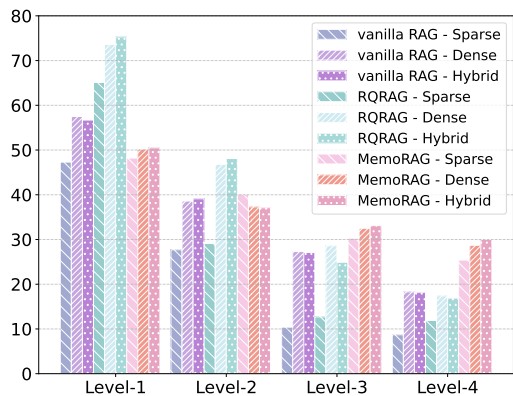

Figure 5: Performance of different retrieval strategies across the four HawkBench levels.

targeting general-purpose or reasoning-intensive tasks. Future research may explore adaptive retrieval modules that dynamically select the most suitable retrieval strategy based on task characteristics.

### 3.3   Key Insights

**Current RAG methods Lack Resilience.** Current RAG methods tend to be optimized for specific types of information-seeking tasks (e.g., fact retrieval or rationale generation). However, this specialization leads to a lack of overall resilience across a broader range of tasks. While empirical analyses provide heuristics to guide method selection for particular tasks, we still lack a systematic,

adaptable solution that can handle diverse tasks with varying requirements. This gap emphasizes the need for developing RAG systems that can dynamically adjust to different information-seeking challenges, moving beyond task-specific optimizations toward a more generalized framework.

**Global Awareness: Construction and Utilization Challenges.** Global awareness is essential for tasks that require the integration of information from multiple sources. However, current global RAG methods struggle with efficiently building and fully leveraging this awareness. While methods such as GraphRAG (which uses graph construction) and memory-based approaches show promise, their reliance on inefficient global intermediate construction processes (e.g., building graphs or memory stores) remains a major bottleneck. For example, graph construction can take tens of minutes, making it impractical for real-time use. Optimizing these construction processes could make these systems more viable. Additionally, there is a need for research into how to best utilize global intermediates (e.g., graphs, memory caches) to improve retrieval and reasoning. Exploring efficient ways to construct and use these intermediates is an important direction for future work.

**Dynamic Task Understanding and Adaptive Query Interpretation.** As information-seeking tasks become more complex, the need for dynamic task understanding and adaptive query interpretation becomes increasingly important. A one-size-fits-all solution is not feasible; instead, RAG systems must integrate decision-making mechanisms that allow them to dynamically adjust how they access (referencing) and utilize (reasoning) knowledge. By understanding the task context and adapting the retrieval strategy accordingly, RAG systems can more effectively address a wider range of queries. This adaptability would significantly enhance the robustness and efficiency of RAG methods, enabling them to handle varying complexities and task types more effectively.

**The Potential of Agentic Information-Seeking Systems.** Looking ahead, agentic information-seeking systems—designed to autonomously navigate knowledge acquisition—offer a compelling direction for the future of AI. By integrating retrieval, reasoning, and synthesis, these systems can perform complex tasks such as literature reviews, report writing, or exploratory research. Recent developments like OpenAI's Deep Research exemplify this trend, signaling a shift toward AI agents that not only assist but independently manage knowledge-intensive workflows. As these systems mature, they hold the potential to reshape how we interact with and generate information, making them a key area for future investigation and innovation.

# 4  Related Work

**RAG Methods**  RAG was introduced by Lewis et al. [2020] to enhance language models' ability to handle knowledge-intensive tasks by providing relevant context through retrieval. Research in RAG has focused on two main areas: (1) improving retrieval quality to set an upper bound for generation accuracy [Qian et al., 2024, Gao et al., 2024], and (2) optimizing the use of retrieved passages for relevance and accessibility during generation [Jiang et al., 2023, Zhao et al., 2024b].

The integration of RAG with LLMs has gained momentum, especially in knowledge-intensive applications [Shuster et al., 2021]. As a result, there is increasing demand for more generalized RAG systems capable of handling a wider range of tasks, including those beyond factoid queries [Zhao et al., 2024a]. However, traditional RAG pipelines face challenges in addressing complex tasks with implicit information needs, often failing to provide sufficient context for accurate generation [Gao et al., 2024, Zhao et al., 2024a]. Recent advances have aimed to expand RAG's applicability. For example, *GraphRAG* [Edge et al., 2024] and *HippoRAG* [Jimenez Gutierrez et al., 2024] introduce knowledge graphs to facilitate retrieval and enhance global awareness. Agent-based approaches, such as *ActiveRAG* [Xu et al., 2024, Yoon et al., 2024], plan information access and utilization via agents.

**RAG Benchmarking**  As RAG systems are increasingly adopted, the need for comprehensive evaluation benchmarks has become evident. Early benchmarks, such as KILT [Petroni et al., 2021], primarily focused on task-specific aspects like single-hop and multi-hop reasoning, as well as factoid queries. Recently, new benchmarks have been developed to address specialized tasks and domains. For example, MultiHop-RAG evaluates multi-hop tasks [Tang and Yang, 2024], LegalBench-RAG focuses on the legal domain [Guha et al., 2023], CRAG offers a comprehensive evaluation framework for factoid question answering tasks, and RAGBench is designed to assess the explainability of RAG systems [Friel et al., 2024]. While these benchmarks provide insights into various facets of RAG

performance, they lack a comprehensive framework to evaluate the resilience of RAG systems when faced with diverse information-seeking needs, particularly for stratified queries [Zhao et al., 2024a].

## 5 Conclusion

In this paper, we introduce HawkBench, a comprehensive framework designed to evaluate the resilience of RAG systems across diverse information-seeking tasks. HawkBench is distinguished by its systematic task stratification, multi-domain corpora, and high-quality annotations, making it an robust tool for assessing the resilience of RAG methods. Our evaluation of representative RAG methods reveals that while current RAG systems are often optimized for specific tasks, they lack resilience across general tasks. This highlights the need for dynamic task strategies that integrate decision-making, query interpretation, and global knowledge utilization to enhance the generalizability of RAG systems. HawkBench serves as a critical resource for advancing the development of resilient, versatile RAG systems capable of addressing a wide range of real-world user needs.

## Acknowledgement

This work was supported by National Natural Science Foundation of China No. 62502049.

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

## A  Implementation Details

In our evaluation of baseline methods on HawkRAG, we use BGE-M3 [Chen et al., 2023] as the retriever for vanilla RAG, RQ-RAG, HyDE, and MemoRAG, setting the hit number to 5. For methods that segment long contexts into chunks, we utilize the semantic-text-splitter tool, limiting chunks to a maximum of 512 tokens. MemoRAG employs the officially released memorag-qwen2-7b-inst as its memory model. For GraphRAG, we leverage GPT-4o for graph construction and use the retrieved context for generation. All baseline methods adopt Qwen-2.5-7B-instruct-128K as the generator.

HawkRAG's raw texts are sourced from books-3-textbooks, legal contracts, arXiv papers, and financial reports. During annotation, the annotator would select either GPT-4o or DeepSeek-v3 as the assisting agent. Our annotation system, illustrated in Figure 6, is implemented using Streamlit. The statistic details of HawkBench are presented in Table 3. In Table 5, we present the full results of the main experiments.

All experiments were conducted on a server equipped with 8 NVIDIA A800-80G GPUs.

Table 3: Statistical Information of HawkBench. The symbols $\langle|\mathcal{C}|\rangle$, $\langle|\mathcal{Q}|\rangle$, and $\langle|\mathcal{A}|\rangle$ represent the average lengths of the context, query, and answer, respectively.

| Dataset | Num | $\langle|\mathcal{C}|\rangle$ | Num | $\langle|\mathcal{Q}|\rangle$ | $\langle|\mathcal{A}|\rangle$ | Num | $\langle|\mathcal{Q}|\rangle$ | $\langle|\mathcal{A}|\rangle$ | Num | $\langle|\mathcal{Q}|\rangle$ | $\langle|\mathcal{A}|\rangle$ | Num | $\langle|\mathcal{Q}|\rangle$ | $\langle|\mathcal{A}|\rangle$ |
| --- | --- | --- | --- | --- | --- | --- | --- | --- | --- | --- | --- | --- | --- | --- |
| | | | | LEVEL-1 | | | LEVEL-2 | | | LEVEL-3 | | | LEVEL-4 | |
| TECHNOLOGY | 200 | 144803.0 | 50 | 15.8 | 5.1 | 50 | 57.7 | 14.1 | 50 | 25.3 | 96.4 | 50 | 26.3 | 42.0 |
| NOVEL | 200 | 166960.2 | 50 | 14.2 | 6.8 | 50 | 51.6 | 19.0 | 50 | 28.2 | 121.5 | 50 | 31.1 | 63.5 |
| ART | 200 | 115591.8 | 50 | 17.0 | 6.9 | 50 | 53.6 | 14.8 | 50 | 27.0 | 125.2 | 50 | 34.4 | 87.7 |
| HUMANITIES | 200 | 152600.3 | 50 | 16.8 | 6.9 | 50 | 56.1 | 26.6 | 50 | 29.1 | 134.1 | 50 | 33.6 | 72.3 |
| PAPER | 200 | 41702.0 | 50 | 18.2 | 9.5 | 50 | 75.7 | 17.1 | 50 | 34.0 | 101.0 | 50 | 28.6 | 40.3 |
| SCIENCE | 200 | 143517.0 | 50 | 16.3 | 7.6 | 50 | 54.3 | 15.3 | 50 | 26.8 | 109.2 | 50 | 29.0 | 47.9 |
| FINANCE | 200 | 37364.6 | 50 | 17.2 | 10.5 | 50 | 62.6 | 12.5 | 50 | 27.0 | 105.6 | 50 | 28.0 | 65.0 |
| LEGAL | 200 | 49331.1 | 50 | 19.3 | 11.9 | 50 | 53.0 | 21.0 | 50 | 27.2 | 113.0 | 50 | 27.0 | 46.7 |
| Total | 1600 | 106483.7 | 400 | 16.8 | 8.2 | 400 | 58.1 | 17.5 | 400 | 28.1 | 113.3 | 400 | 29.7 | 58.2 |

## B  Limitations

This paper focuses on constructing a benchmark, HawkBench, to evaluate the resilience of RAG methods across stratified tasks. While the benchmark provides a comprehensive framework, there are several limitations to consider. First, dataset bias may arise during the curation process, as the raw data are collected from multiple domains. This diversity, while beneficial, may inadvertently introduce biases that could affect the generalizability of the results. Additionally, during the annotation process, both the assisting LLMs and human annotators may introduce errors, which could impact the overall evaluation quality. Although we strive for thoroughness in evaluating task and domain diversity, HawkBench's size, while reasonable, may not cover all professional knowledge-intensive domains or task types.

Furthermore, while we conduct comprehensive experiments using HawkBench, it is not feasible to test all available RAG methods, alternative retrievers, or LLMs on this benchmark. We selected representative methods and models that are expected to provide generalizable findings, but this selection does not encompass the full range of possible approaches. Additionally, we did not evaluate commercial RAG solutions in this study, as these systems are typically closed-sourced and subject to changes over time, making them challenging to incorporate into a static benchmark evaluation.

## C  Public Data and Model Memorization vs. Genuine Retrieval

Most publicly available web data, including domain-specific corpora, are likely included in the pre-training corpus of today's large language models. This challenge is shared by nearly all modern NLP benchmarks.

Nevertheless, benchmarks built on such corpora remain meaningful for several reasons. First, seeing a text during pre-training does not guarantee full memorization, nor does it ensure accurate answers for queries requiring complex reasoning or synthesis. Our benchmark's query–answer pairs are manually

| Domain & Level | gemini-2.5-flash | gpt-4o-mini | gpt-4.1 | Best RAG (Qwen-2.5 7B) |
|---|---|---|---|---|
| Legal-Level 1 | 13.6 | 10.5 | 15.3 | **79.2** |
| Legal-Level 2 | 25.2 | 20.6 | 30.0 | **45.9** |
| Legal-Level 3 | 15.8 | 20.7 | 22.0 | **32.7** |
| Legal-Level 4 | 12.4 | 13.2 | 14.3 | **17.9** |
| Finance-Level 1 | 13.6 | 12.3 | 22.3 | **79.5** |
| Finance-Level 2 | 25.2 | 13.3 | 23.1 | **54.7** |
| Finance-Level 3 | 13.8 | 17.8 | 20.2 | **30.8** |
| Finance-Level 4 | 13.4 | 15.8 | 18.9 | **20.4** |
| Science-Level 1 | 13.6 | 12.2 | 13.2 | **45.1** |
| Science-Level 2 | 25.1 | 17.8 | 21.2 | **33.8** |
| Science-Level 3 | 15.7 | 21.4 | 20.1 | **27.3** |
| Science-Level 4 | 13.4 | 17.2 | 17.4 | **20.4** |

Table 4: Comparison of commercial LLM APIs with the best-performing RAG system (Qwen-2.5 7B). RAG substantially outperforms LLMs in factoid queries (Level 1 and 2), while LLMs remain competitive in higher-level reasoning tasks (Level 3 and 4).

annotated to capture nuanced, multi-step information-seeking behaviors that go well beyond simple fact recall.

Second, to mitigate concerns regarding memorization versus genuine retrieval, we include evaluations using several strong commercial LLM APIs. By comparing the performance of a range of models on our benchmark, we can better assess the extent to which retrieval-augmented reasoning (rather than memorization) contributes to success. Specifically, we compare RAG methods with three leading commercial LLMs. The results are shown in Table 4.

The results demonstrate that while strong LLMs have memorized substantial information from public corpora, they still lag behind retrieval-augmented methods in overall performance. Notably, for factoid queries at Level 1 and Level 2, RAG methods outperform strong LLMs by a large margin, suggesting that even with exposure to the underlying texts during pre-training, LLMs cannot reliably recall fine-grained factual details. For Level 3 and Level 4 tasks, which require summarizing broad content or synthesizing information, strong LLMs perform comparably to RAG methods, as these queries demand less precise retrieval and more general reasoning.

In summary, these results show that **even though portions of HawkBench may have been seen by strong LLMs during pretraining, it remains a robust benchmark for evaluating stratified RAG performance**. Without retrieval, even advanced LLMs such as GPT-4.1 can only solve a small fraction of the tasks, highlighting the necessity of effective retrieval-augmented reasoning. Moreover, these experiments suggest that HawkBench not only provides a comprehensive testbed for RAG evaluation, but also serves as a tool for assessing the factual memorization capabilities of state-of-the-art LLMs.

# D Broader Impact

Our work aims to advance the robustness and generalizability of RAG systems by introducing a comprehensive benchmark, HawkBench, that stratifies tasks based on real-world information-seeking complexity. This can benefit a wide range of applications—such as question answering, legal and financial document analysis, and educational tutoring—by enabling more adaptive and reliable retrieval-augmented language models.

However, improving general-purpose information-seeking systems also raises concerns. These include the risk of propagating misinformation from retrieved content, amplifying biases present in training or retrieval corpora, and enabling misuse in sensitive domains without sufficient oversight. We encourage developers to adopt careful evaluation and safeguards when deploying RAG systems, especially in high-stakes or regulated scenarios.

Ultimately, we hope that HawkBench facilitates more transparent, equitable, and effective development of retrieval-based AI systems, while fostering research into more accountable and context-aware reasoning mechanisms.

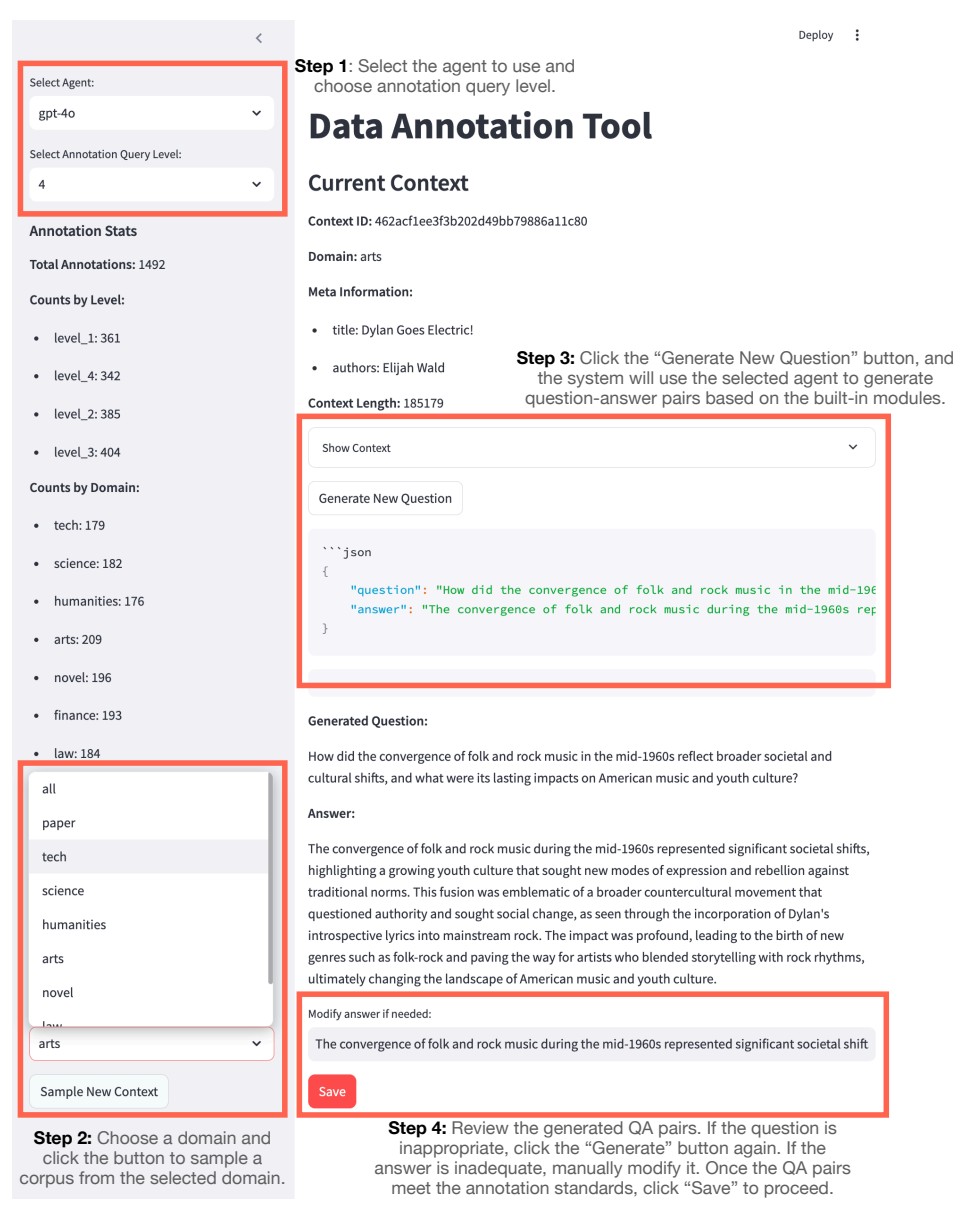

Figure 6: Annotation Interface of HawkBench.

Table 5: Full details of main experimental results.

| Dataset LEVEL-1 | TECH R-L | F1 | NOV R-L | F1 | ART R-L | F1 | HUM R-L | F1 | PAPER R-L | F1 | SCI R-L | F1 | FIN R-L | F1 | LEG R-L | F1 | AVE R-L | F1 |
|---|---|---|---|---|---|---|---|---|---|---|---|---|---|---|---|---|---|---|
| Top-1 | 47.8 | 52.6 | 39.9 | 42.4 | 39.5 | 46.2 | 25.2 | 27.9 | 30.8 | 32.7 | 29.3 | 44.1 | 87.2 | 88.5 | 38.5 | 39.1 | 42.3 | 46.7 |
| Top-5 | 59.0 | 66.2 | 47.0 | 55.1 | 44.2 | 54.0 | 23.5 | 28.8 | 58.2 | 64.0 | 44.6 | 57.1 | 80.3 | 83.9 | 50.4 | 50.9 | 50.9 | 57.5 |
| Top-10 | 69.0 | 74.4 | 58.3 | 70.7 | 56.8 | 72.2 | 58.6 | 69.4 | 57.5 | 62.1 | 45.6 | 65.4 | 77.6 | 80.9 | 74.1 | 75.0 | 62.2 | 71.3 |
| Top-50 | 12.8 | 12.7 | 33.8 | 35.9 | 25.3 | 29.3 | 14.1 | 16.4 | 27.6 | 28.0 | 20.4 | 23.3 | 43.5 | 44.2 | 35.3 | 37.1 | 26.6 | 28.4 |
| LLM | 7.5 | 6.8 | 6.3 | 6.1 | 7.1 | 6.9 | 6.8 | 5.8 | 20.1 | 20.4 | 8.2 | 9.1 | 24.6 | 24.8 | 23.5 | 23.3 | 13.0 | 12.9 |
| Lingua-2 | 5.0 | 3.5 | 3.8 | 3.0 | 8.9 | 9.1 | 3.0 | 2.1 | 8.2 | 7.4 | 5.9 | 6.2 | 27.0 | 28.2 | 29.8 | 31.4 | 11.4 | 11.4 |
| MInference | 8.0 | 7.1 | 5.3 | 5.1 | 7.2 | 7.1 | 5.3 | 5.1 | 15.7 | 13.9 | 7.2 | 7.7 | 23.3 | 23.2 | 19.7 | 19.7 | 11.5 | 11.1 |
| HyDE | 71.2 | 78.2 | 57.2 | 69.4 | 56.5 | 71.4 | 62.5 | 72.9 | 63.6 | 66.7 | 47.3 | 67.5 | 79.5 | 82.1 | 77.8 | 79.5 | 64.4 | 73.5 |
| RQRAG | 72.0 | 78.2 | 55.4 | 68.3 | 59.0 | 74.2 | 57.5 | 70.7 | 65.7 | 67.6 | 45.1 | 66.4 | 80.9 | 83.9 | 78.2 | 79.2 | 64.2 | 73.6 |
| MemoRAG | 46.9 | 51.4 | 29.6 | 34.9 | 35.1 | 46.0 | 48.5 | 55.0 | 32.9 | 35.1 | 35.1 | 44.6 | 65.4 | 68.2 | 65.1 | 66.5 | 44.8 | 50.2 |
| GraphRAG | 58.7 | 66.8 | 52.5 | 58.0 | 44.3 | 55.9 | 48.1 | 57.0 | 28.8 | 34.9 | 39.6 | 58.5 | 67.7 | 72.7 | 54.3 | 55.3 | 49.3 | 57.4 |

| Dataset LEVEL-2 | TECH R-L | F1 | NOV R-L | F1 | ART R-L | F1 | HUM R-L | F1 | PAPER R-L | F1 | SCI R-L | F1 | FIN R-L | F1 | LEG R-L | F1 | AVE R-L | F1 |
|---|---|---|---|---|---|---|---|---|---|---|---|---|---|---|---|---|---|---|
| Top-1 | 35.6 | 40.3 | 20.1 | 22.5 | 27.4 | 33.4 | 26.7 | 32.2 | 32.8 | 38.0 | 22.3 | 25.5 | 40.5 | 42.5 | 33.1 | 37.1 | 29.8 | 34.0 |
| Top-5 | 37.1 | 41.7 | 28.9 | 35.8 | 29.8 | 34.8 | 30.8 | 36.6 | 40.9 | 45.6 | 23.0 | 25.1 | 44.0 | 47.1 | 37.1 | 41.7 | 34.0 | 38.6 |
| Top-10 | 39.1 | 45.0 | 40.9 | 49.3 | 31.9 | 40.4 | 35.0 | 39.6 | 40.9 | 43.9 | 28.9 | 31.7 | 51.9 | 54.2 | 45.4 | 51.5 | 39.3 | 44.4 |
| Top-50 | 20.1 | 21.6 | 25.2 | 26.8 | 18.9 | 21.3 | 22.7 | 25.4 | 21.0 | 19.2 | 16.8 | 18.4 | 23.0 | 22.0 | 25.3 | 27.5 | 21.6 | 22.8 |
| LLM | 10.1 | 8.6 | 11.4 | 10.6 | 11.5 | 11.7 | 14.8 | 12.8 | 17.1 | 14.2 | 9.8 | 8.8 | 10.4 | 8.5 | 18.4 | 16.6 | 12.9 | 11.5 |
| Lingua-2 | 10.2 | 8.9 | 10.1 | 9.3 | 8.9 | 9.8 | 13.2 | 12.6 | 14.6 | 12.2 | 9.8 | 8.9 | 13.2 | 12.1 | 17.5 | 17.5 | 12.2 | 11.4 |
| MInference | 9.7 | 8.1 | 11.8 | 11.6 | 11.0 | 10.7 | 13.9 | 12.3 | 15.9 | 13.6 | 9.9 | 8.7 | 10.7 | 8.8 | 17.6 | 15.7 | 12.6 | 11.2 |
| HyDE | 45.3 | 48.6 | 39.7 | 46.7 | 33.5 | 39.3 | 33.2 | 40.2 | 39.0 | 43.4 | 30.2 | 32.3 | 54.3 | 55.6 | 45.1 | 49.9 | 40.0 | 44.5 |
| RQRAG | 44.4 | 48.0 | 38.5 | 46.2 | 36.5 | 45.4 | 33.3 | 40.9 | 41.5 | 47.5 | 33.8 | 37.2 | 54.7 | 58.6 | 45.9 | 50.2 | 41.1 | 46.8 |
| MemoRAG | 33.0 | 37.9 | 26.2 | 30.4 | 30.2 | 36.0 | 31.1 | 35.3 | 38.8 | 42.0 | 24.7 | 26.1 | 46.6 | 48.1 | 39.2 | 42.6 | 33.7 | 37.3 |
| GraphRAG | 34.8 | 38.9 | 35.9 | 40.7 | 28.9 | 30.9 | 33.5 | 38.7 | 31.7 | 31.0 | 25.0 | 27.4 | 45.9 | 48.6 | 36.5 | 39.4 | 34.0 | 37.0 |

| Dataset LEVEL-3 | TECH R-L | S-F1 | NOV R-L | S-F1 | ART R-L | S-F1 | HUM R-L | S-F1 | PAPER R-L | S-F1 | SCI R-L | S-F1 | FIN R-L | S-F1 | LEG R-L | S-F1 | AVE R-L | S-F1 |
|---|---|---|---|---|---|---|---|---|---|---|---|---|---|---|---|---|---|---|
| Top-1 | 15.5 | 26.9 | 12.4 | 14.6 | 12.3 | 26.0 | 10.1 | 19.9 | 20.3 | 23.5 | 17.8 | 26.0 | 12.6 | 13.7 | 19.2 | 23.2 | 15.0 | 21.7 |
| Top-5 | 15.5 | 27.5 | 15.7 | 22.2 | 14.4 | 30.0 | 16.4 | 22.1 | 22.9 | 27.7 | 19.0 | 34.9 | 17.2 | 22.5 | 22.3 | 31.4 | 17.9 | 27.3 |
| Top-10 | 22.3 | 33.3 | 20.4 | 22.8 | 18.4 | 35.6 | 19.2 | 28.1 | 30.3 | 44.9 | 24.2 | 43.6 | 22.4 | 26.3 | 27.9 | 33.1 | 23.1 | 33.5 |
| Top-50 | 18.9 | 25.5 | 19.8 | 23.3 | 19.3 | 30.5 | 24.5 | 29.7 | 26.5 | 30.7 | 23.9 | 37.2 | 27.1 | 31.2 | 26.7 | 35.4 | 23.3 | 30.4 |
| LLM | 23.8 | 20.1 | 23.3 | 17.4 | 23.2 | 19.2 | 23.7 | 23.8 | 30.3 | 34.8 | 24.6 | 26.1 | 30.2 | 30.9 | 30.3 | 26.5 | 26.2 | 24.9 |
| Lingua-2 | 19.8 | 16.3 | 21.9 | 17.4 | 19.9 | 18.4 | 20.6 | 18.5 | 26.7 | 31.8 | 22.0 | 19.5 | 30.8 | 35.9 | 28.4 | 33.9 | 23.7 | 23.9 |
| MInference | 23.3 | 19.8 | 23.2 | 16.8 | 22.1 | 20.2 | 23.5 | 23.8 | 29.9 | 34.8 | 24.6 | 25.8 | 29.4 | 25.0 | 28.8 | 27.3 | 25.6 | 24.2 |
| HyDE | 17.7 | 34.6 | 16.0 | 14.1 | 16.7 | 33.5 | 16.4 | 21.6 | 26.1 | 35.7 | 21.6 | 36.7 | 16.6 | 24.3 | 24.1 | 23.5 | 19.4 | 28.0 |
| RQRAG | 17.6 | 31.7 | 15.7 | 23.1 | 17.8 | 32.4 | 16.3 | 20.9 | 25.3 | 32.9 | 20.8 | 37.8 | 17.5 | 23.5 | 26.4 | 26.9 | 19.7 | 28.6 |
| MemoRAG | 23.2 | 33.4 | 24.5 | 25.7 | 25.1 | 29.8 | 26.0 | 29.8 | 32.6 | 44.5 | 27.3 | 42.0 | 26.9 | 33.4 | 32.7 | 34.3 | 27.3 | 34.1 |
| GraphRAG | 22.1 | 31.6 | 23.8 | 29.0 | 22.2 | 33.2 | 24.6 | 23.9 | 31.4 | 44.0 | 27.2 | 42.7 | 24.3 | 29.2 | 26.7 | 26.1 | 25.3 | 32.5 |

| Dataset LEVEL-4 | TECH R-L | S-F1 | NOV R-L | S-F1 | ART R-L | S-F1 | HUM R-L | S-F1 | PAPER R-L | S-F1 | SCI R-L | S-F1 | FIN R-L | S-F1 | LEG R-L | S-F1 | AVE R-L | S-F1 |
|---|---|---|---|---|---|---|---|---|---|---|---|---|---|---|---|---|---|---|
| Top-1 | 16.3 | 24.5 | 13.1 | 20.4 | 14.9 | 9.0 | 15.9 | 13.8 | 17.7 | 17.4 | 14.7 | 15.6 | 13.0 | 13.0 | 11.8 | 16.5 | 14.7 | 16.3 |
| Top-5 | 16.8 | 23.4 | 14.4 | 26.2 | 17.7 | 16.1 | 15.3 | 16.6 | 16.9 | 15.6 | 17.7 | 21.5 | 11.8 | 9.9 | 11.7 | 16.9 | 15.3 | 18.3 |
| Top-10 | 21.1 | 40.2 | 17.4 | 22.7 | 20.3 | 17.1 | 18.4 | 22.9 | 20.5 | 16.1 | 18.0 | 19.3 | 16.0 | 19.9 | 13.8 | 8.3 | 18.2 | 20.8 |
| Top-50 | 17.8 | 41.4 | 16.7 | 33.9 | 17.3 | 32.7 | 17.5 | 34.1 | 17.5 | 28.0 | 15.9 | 41.7 | 15.9 | 31.6 | 16.7 | 34.1 | 16.9 | 34.7 |
| LLM | 16.2 | 35.0 | 17.4 | 37.3 | 16.8 | 34.4 | 17.2 | 29.6 | 17.1 | 31.4 | 15.2 | 32.2 | 19.6 | 35.2 | 15.6 | 30.1 | 16.9 | 33.2 |
| Lingua-2 | 13.9 | 32.2 | 14.1 | 23.3 | 15.0 | 24.9 | 13.8 | 10.5 | 17.5 | 27.0 | 12.9 | 22.3 | 20.4 | 35.8 | 15.8 | 25.1 | 15.4 | 25.2 |
| MInference | 16.2 | 37.3 | 18.6 | 34.5 | 16.9 | 37.0 | 17.7 | 28.0 | 17.1 | 32.4 | 15.7 | 28.1 | 18.8 | 35.6 | 15.5 | 33.6 | 17.1 | 33.3 |
| HyDE | 16.8 | 25.6 | 14.8 | 20.0 | 16.7 | 14.2 | 15.5 | 16.9 | 15.2 | 16.7 | 19.4 | 20.5 | 13.7 | 18.6 | 12.6 | 15.2 | 15.6 | 18.4 |
| RQRAG | 16.0 | 22.1 | 14.8 | 17.8 | 17.6 | 15.8 | 16.0 | 17.0 | 15.3 | 17.9 | 17.7 | 24.0 | 13.2 | 13.1 | 12.8 | 11.1 | 15.4 | 17.4 |
| MemoRAG | 17.7 | 43.8 | 20.0 | 44.1 | 19.8 | 37.2 | 19.7 | 37.8 | 20.4 | 26.1 | 16.9 | 36.3 | 19.8 | 30.1 | 17.9 | 24.2 | 19.0 | 35.0 |
| GraphRAG | 20.7 | 37.3 | 21.1 | 37.5 | 22.7 | 34.4 | 22.4 | 31.4 | 23.7 | 21.5 | 20.4 | 27.4 | 19.2 | 20.2 | 15.1 | 19.5 | 20.6 | 28.7 |

