# OpenReview forum: "HawkBench: Investigating Resilience of RAG Methods on Stratified Information-Seeking Tasks"
_NeurIPS.cc/2025/Datasets_and_Benchmarks_Track — NeurIPS 2025 Datasets and Benchmarks Track spotlight_

### Official Review · Reviewer_xziF · 2025-06-30

**Ethics Flags:** Data privacy, copyright, and consent
**Rating:** 5
**Confidence:** 3

**Summary:**

This paper introduces HawkBench, a new human-labeled, multi-domain benchmark to assess RAG performance across categorized task types. It includes 1,600 test samples evenly distributed across domains and task types. Evaluation of representative RAG methods with this benchmark in answer quality and response latency highlights the need for dynamic task strategies that integrate decision-making, query interpretation, and global knowledge understanding to improve RAG generalizability.

**Additional Feedback:**

- It's not clear enough whether Figure 4 corresponds to the results for Naive RAG or an aggregation over multiple RAG approaches.
- It seems that some usage of \citet should be replaced by \citep, e.g., the ones in Section 4.

**Dataset Code Accessibility:**

Yes

**Dataset Code Comments:**

Dataset is available on Huggingface and code repo is available on GitHub. The README file in the code repo details the instructions for setting up the environment and running the scripts.

**Ethical Comments:**

The guidelines for checklist question 12 are not fully complied with, e.g., lack of license including for existing assets.

**Ethical Considerations:**

Yes, there are ethics concerns that require attention by the authors

**Final Justification:**

The issues I raised earlier have been mostly resolved. Specifically:

- The authors clarified the connection between the adopted query stratification and prior categorizations, detailing the benefit realized by the new stratification.
- The authors explained in detail the proposed S-F1 metric and its connection to existing LLM-as-a-Judge practices.
- The authors provided additional empirical results with another LLM to verify the robustness of the empirical observations across LLMs.
- The authors clarified about data privacy and copyright.

**Limitations Weaknesses:**

- **W1.** Query stratification in Section 2.2 reads like a re-branding and/or refinement of existing tasks already employed for RAG evaluation -- multi-hop question answering and query-focused summarization. Even though this was proposed by an existing paper, more detailed discussions are expected to justify its adoption in terms of:
    1. the conceptual relationship between the proposed stratification and existing task classification
    2. the conceptual benefit of the stratification over existing task classification
    3. the gain in empirical insights arising from the proposed task stratification
- **W2.** The evaluation is limited in terms of the LLM generator (Qwen2.5-7B-instruct).
- **W3.** There lack discussions of the proposed S-F1 metric with respect to existing LLM-as-a-judge practices, both in concept and empirical qualitative analyses.
- **W4.** Insights like no one-size-fits-all approaches and the need for global knowledge understanding have been discussed more or less in prior or concurrent works. Some discussions with respect to those prior works are expected to better understand the contributions. [1-4] are some examples.

[1] RAPTOR: Recursive Abstractive Processing for Tree-Organized Retrieval.

[2] From Local to Global: A Graph RAG Approach to Query-Focused Summarization

[3] From RAG to Memory: Non-Parametric Continual Learning for Large Language Models

[4] RAG vs. GraphRAG: A Systematic Evaluation and Key Insights

**Strengths Contributions:**

- **S1.** The paper is overall easy to follow.
- **S2.** HawkBench considers a diverse range of sources -- including professional textbooks, academic papers, financial reports, legal contracts, and novels.
- **S3.** Human annotation is considered for quality control.
- **S4.** Diverse RAG and long-context LLM approaches are considered for evaluation.
- **S5.** A new metric S-F1 for evaluating answers to challenging rationale queries based on LLM-as-a-judge.
- **S6.** Empirical studies disclose how different domains lead to distinct observations, which justify the need of employing diverse domains for evaluation.
- **S7.** Ablation studies are conducted to disclose the impact of retriever choice (sparse, dense, and hybrid).

---

> ### Author Rebuttal · Authors · 2025-07-30
>
> Thank you for your insightful review and constructive feedback. We appreciate your recognition of HawkBench’s diverse sources, human annotation pipeline, comprehensive evaluation, and introduction of the S-F1 metric. Below, we address your comments in detail.
>
> ---
>
> ### 1. Query Stratification: Relationship, Benefit, and Insights
>
> Thank you for your thoughtful question regarding query stratification. We agree that our framework is related to prior categorizations (e.g., multi-hop QA, query-focused summarization), but our design goes beyond a mere re-branding. Specifically:
>
> **a) Conceptual relationship:**
> Our stratification draws on and extends the principles behind factoid QA, multi-hop QA, and query-focused summarization. We unify these diverse task types into four well-defined levels based on two orthogonal axes: (i) explicitness of query intent (explicit vs. implicit) and (ii) required reasoning complexity (factoid vs. rationale). This provides a more interpretable and consistent taxonomy that clarifies and connects previously fragmented task definitions (see Section 2.2).
>
> **b) Conceptual benefit:**
> Unlike previous benchmarks—which evaluate each task on different corpora, hindering fair comparison—HawkBench stratifies all query types within the same domain corpus and uses consistent annotation. This design ensures direct comparability between, for example, multi-hop QA and summarization queries, as both are anchored to the same underlying documents. Such consistency mitigates confounding effects of corpus or annotation variation, providing a clearer measurement of model generalization and robustness.
>
> **c) Empirical insights:**
> Our unified benchmark enables systematic analysis of RAG performance across task types in a controlled setting. For instance, we empirically show (Section 3.2, Table 1) that methods optimized for global awareness (e.g., GraphRAG) excel on rationale-based queries but may underperform on factoid tasks. Such comparative insights would be less meaningful if task types were separated across different corpora. Our stratification also allows us to observe trade-offs and transfer effects between reasoning and referencing requirements.
>
> We will further clarify these distinctions and their implications in the revised manuscript.
>
> ---
>
> ### 2. S-F1 Metric vs. Existing LLM-as-a-Judge Practices
>
> Thank you for raising this important point. The key innovation of our S-F1 metric lies in its **sentence-level granularity**. While most LLM-as-a-judge approaches generate a coarse, overall judgment (e.g., win/tie/loss or a single score for the whole answer), we found through human audit that such methods often fail to faithfully reflect nuanced answer quality—particularly for long or rationale-heavy responses. For example, a lengthy answer with mostly correct content but one flawed sentence might be overly penalized by holistic metrics.
>
> In contrast, S-F1 leverages an LLM-as-a-judge at the sentence level:
> - It computes **sentence-level recall** (fraction of reference sentences supported by the prediction) and **precision** (fraction of predicted sentences supported by the reference).
> - This fine-grained evaluation better aligns with human judgments, especially for rationale-based or long-form answers, and provides interpretable feedback on both completeness and correctness.
>
> We will clarify the conceptual advantages and add empirical illustrations of S-F1’s effectiveness in the revised manuscript.
>
> ---
>
> ### 3. Contextualizing “No One-Size-Fits-All” & Global Knowledge Insights
>
> Thank you for this observation. We fully acknowledge that limitations of “one-size-fits-all” approaches and the importance of global knowledge understanding have been discussed in prior and concurrent works. However, previous discussions have typically been qualitative or post-hoc, often to explain specific cases of model underperformance.
>
> Our contribution is to **systematically and empirically validate** these insights within a unified benchmark. By leveraging HawkBench’s controlled query stratification and consistent corpora, we quantitatively demonstrate where and why different RAG methods succeed or struggle across task types and domains. This enables a more rigorous, evidence-based understanding than was previously possible.
>
> We will explicitly discuss these related works and position our unique empirical contributions in the revised manuscript.
>
> ---
>
> ### 4. LLM Generator Diversity
>
> The reason we selected Qwen 2.5-7B as the primary backbone is that some RAG methods evaluated in our study are model-dependent. For example, MInference currently supports only a limited range of model types and does not support all popular open-source LLMs.
>
> In response to your suggestion, we conducted additional experiments using Mistral-7B-Instruct-v0.2 as the backbone model. The results, presented below, show that our main findings remain consistent: the relative strengths and weaknesses of the RAG baselines are similar to those observed with Qwen2.5-7B-instruct.
>
>
> | Method       | Type         | Level-1 Rouge-L | Level-1 F1 | Level-2 Rouge-L | Level-2 F1 | Level-3 Rouge-L | Level-3 S-F1 | Level-4 Rouge-L | Level-4 S-F1 |
> | ------------ | ------------ | --------------- | ---------- | --------------- | ---------- | --------------- | ------------ | --------------- | ------------ |
> | **LLM**      | Long LLM     | 10.1            | 11.1       | 12.2            | 10.5       | _25.0_          | 23.2         | 16.1            | 32.0         |
> | **Lingua-2** | Compression  | 10.8            | 10.9       | 11.8            | 10.3       | 22.7            | 22.9         | 14.9            | 24.1         |
> | **RAG**      | Standard RAG | 44.2            | 53.2       | 30.2            | 33.1       | 16.4            | 25.8         | 14.1            | 16.5         |
> | **HyDE**     | Enhanced RAG | **62.6**        | **74.2**     | _41.1_          | _45.7_     | 18.9            | 28.3         | 14.7            | 17.8         |
> | **RQRAG**    | Enhanced RAG | _62.4_          | _74.1_   | **41.8**        | **47.9**   | 18.3            | 27.6         | 15.2            | 18.0         |
> | **MemoRAG**  | Global RAG   | 43.5            | 49.3       | 31.9            | 35.5       | **29.1**        | **35.0**     | **22.1**        | **34.5**     |
> | **GraphRAG** | Global RAG   | 48.1            | 55.9       | 33.6            | 35.9       | 23.9            | _31.7_       | _21.5_          | _30.0_       |
>
> Please note that since MInference is not compatible with Mistral models, it is excluded from these additional baselines.
>
>
> ---
>
> ### 5. Ethics, Data Privacy, and Consent
>
> Thank you for your attention to data privacy, copyright, and consent. We fully recognize the importance of these ethical considerations.
>
> - As stated in lines 216–220, we have carefully reviewed the licenses of all datasets used. All data sources are publicly available via HuggingFace, and we comply with their respective licenses. Constructing benchmarks based on publicly available web corpora is a widely adopted standard in the research community.
> - To ensure transparency and verification, we provide URLs for all data sources (lines 543–544).
> - If needed, we offer an **alternative approach**: instead of distributing raw text, we provide a script for users to gather and preprocess data from the original sources. This respects data ownership and facilitates reproduction and extension.
> - We do not use these sources for model training, only for evaluation, and do not enable commercial use.
>
> If further clarification or adjustment is needed to fully satisfy ethical checklist requirements, we are happy to add additional details.
>
> ---
>
> ### 6. Additional Feedback
>
> - **Figure 4 Clarity:**
>   Thank you for pointing out the ambiguity. Figure 4 reports results for vanilla RAG with varying Top-k values. This experiment is designed to systematically investigate Top-k selection. We will clarify this in the revision.
>
> - **Citation Format:**
>   We appreciate the feedback regarding citation format and will correct all relevant instances in the revised manuscript.

---

> > ### Comment · Reviewer_xziF · 2025-08-01
> >
> > Thank you for the detailed response. I don't have any other major questions or concerns. I'll adjust my evaluation accordingly.

---

> > > ### Author Response · Authors · 2025-08-04
> > >
> > > Thank you for your detailed feedback. We’re glad our responses have clarified your concerns and appreciate your consideration in updating your evaluation.

---

### Official Review · Reviewer_kENp · 2025-07-01

**Rating:** 5
**Confidence:** 3

**Summary:**

This paper introduces HawkBench, a new benchmark designed to test the resilience of Retrieval-Augmented Generation (RAG) methods across different real-world information-seeking tasks and domains. HawkBench uses 1,600 human-labeled queries, organized by task and domain type, to more fairly evaluate RAG methods.

**Dataset Code Accessibility:**

Yes

**Ethical Considerations:**

No, there are no or only very minor ethics concerns

**Final Justification:**

I think my concerns have been solved. Thanks.

**Limitations Weaknesses:**

1. The use of mainly overlap-based metrics may not fully reflect answer quality for complex tasks; LLM-based precision and recall could provide more accurate evaluation. For example, extracting facts/ claims from ground-truth and response then comparing to each other might be more useful.
2. Commercial models like GPT-4, Claude, or Gemini often lead state-of-the-art performance. ` Qwen2.5-7B-instruct` seems to be not enough to show the resilience across levels in the main results.
3. Does “229 contexts” refer to 229 individual documents? If so, could the authors please provide more detailed statistics—such as the average and maximum document length (in tokens or words), as well as the typical number of pages? This would help assess the complexity of the data and the challenges involved in scaling the database. Without such information, it's difficult to gauge whether the setup represents a realistic and demanding retrieval scenario.

**Strengths Contributions:**

1. HawkBench supports a wide range of information-seeking tasks, not just simple factoid questions.
2. This paper gives clear guidance on weaknesses of current RAG methods and show where improvements are needed.
3. Multi-domain coverage, human-labeled data, and systematic annotation process make the dataset robust and high-quality.  In contrast, many existing multi-hop RAG evaluation datasets often rely on artificially constructed queries—such as forced 3-hop questions—that may increase difficulty but lack realism or alignment with natural user intent. I appreciate that the questions in this dataset are both fluent and grounded in genuine human information needs, which enhances its practical value for real-world applications.
4. The paper is well-organized, logically structured, and clear to read.
5. I really liked the range of baselines included in the paper—it gave me a much better sense of which type of RAG setup works best in different scenarios. It’s helpful to see side-by-side comparisons like this because it makes it easier to choose the right approach depending on the task.

---

> ### Author Rebuttal · Authors · 2025-07-30
>
> Thank you for your thoughtful and detailed review. We appreciate your recognition of HawkBench’s realistic human-labeled queries, multi-domain coverage, robust annotation pipeline, clear comparative analysis, and overall clarity. We address your specific suggestions and concerns below:
>
> ---
>
> ### 1. Metrics for Complex Tasks: Beyond Overlap-Based Evaluation
>
> We fully agree that overlap-based metrics may not adequately capture answer quality for complex information-seeking tasks. In our work, lexical metrics (such as F1-score and Rouge-L) are applied mainly to Level-1 and Level-2 questions, where answers are short and factual.
>
> For more complex, rationale-based tasks (Levels 3 and 4), we introduce our carefully designed **S-F1** metric. Unlike simple LLM-as-judge scores—which tend to be holistic and sometimes coarse—S-F1 operates at the sentence level and computes both precision and recall: a strong LLM judge determines whether each sentence in the ground-truth answer is semantically supported by the model’s response, and vice versa. Aggregating these judgments yields a more granular, interpretable, and reliable assessment of completeness and accuracy.
>
> This design makes S-F1 significantly more robust than purely overlap-based or single-score LLM-judged metrics, especially for evaluating reasoning, synthesis, and nuanced claim matching. We will clarify the computation and advantages of S-F1 in the revised manuscript.
>
> ---
>
> ### 2. Evaluation with Stronger Commercial LLMs
>
> Thank you for pointing out the need to assess resilience against state-of-the-art commercial models such as GPT-4 and Gemini. In response, we compare RAG methods with three leading commercial LLM APIs. The results show that, although strong LLMs have memorized substantial information from public corpora, they still lag behind retrieval-augmented methods overall.
>
> | Domain & Level | gemini-2.5-flash | gpt-4o-mini | gpt-4.1 | Best RAG (using Qwen-2.5 7B)      |
> | ----- | ---------------- | ----------- | -------------------- | -------- |
> | Legal-Level 1 | 13.6             | 10.5        | 15.3                 | **79.2** |
> | Legal-Level 2 | 25.2             | 20.6        | 30.0                    | **45.9** |
> | Legal-Level 3 | 15.8             | 20.7        | 22.0                  | **32.7** |
> | Legal-Level 4 | 12.4             | 13.2        | 14.3                   | **17.9** |
> | Finance-Level 1 | 13.6             | 12.3        | 22.3                   | **79.5** |
> | Finance-Level 2 | 25.2             | 13.3        | 23.1                    | **54.7** |
> | Finance-Level 3 | 13.8             | 17.8        | 20.2                    | **30.8** |
> | Finance-Level 4 | 13.4             | 15.8       | 18.9                    | **20.4** |
> | Science -Level 1 | 13.6             | 12.2        | 13.2                    | **45.1** |
> | Science -Level 2 | 25.1             | 17.8        | 21.2                     | **33.8** |
> | Science -Level 3 | 15.7             | 21.4        | 20.1                  | **27.3** |
> | Science -Level 4 | 13.4             | 17.2      | 17.4              | **20.4** |
>
> Notably, for factoid queries, RAG methods outperform strong LLMs by a large margin—suggesting that even with exposure to the relevant texts during pretraining, LLMs cannot reliably recall detailed factual information. For Level-3 and Level-4 tasks, which require summarizing broad content or synthesizing information, strong LLMs perform comparably to RAG methods, as these queries require less precise retrieval and more general reasoning.
>
> ---
>
> ### 3. Context Statistics and Retrieval Scenario Complexity
>
> Thank you for your question regarding the document context statistics. In line 216 and Table 3 (Appendix), we already present detailed statistics for HawkBench, including the average context length, query length, and answer length for each domain. To clarify, the “229 contexts” refers to 229 individual documents.
>
> For additional transparency, we also provide the maximum document length for each domain dataset. These statistics help assess the complexity of our retrieval scenario and the challenge of scaling to large, realistic document databases. The information shows as below:
>
> | Domain     | Average Length | Max Length |
> |------------|---------------|------------|
> | Technology | 144,803.0     | 437,070    |
> | Science    | 143,517.0     | 491,066    |
> | Humanity   | 152,600.3     | 688,074    |
> | Arts       | 115,591.8     | 381,043    |
> | Novel      | 166,960.2     | 408,969    |
> | Finance    | 37,364.6      | 68,401     |
> | Legal      | 49,331.1      | 74,192     |
> | Paper      | 41,702.0      | 64,521     |
>
>  We will ensure all such details are clearly presented and discussed in the revised manuscript.

---

> > ### Comment · Reviewer_kENp · 2025-08-04
> >
> > I think my concerns have been solved. Thanks.

---

> > > ### Author Response · Authors · 2025-08-04
> > >
> > > Thank you for your feedback. We’re glad to hear your concerns have been addressed. We appreciate your time and consideration!

---

### Official Review · Reviewer_kbUp · 2025-07-02

**Rating:** 4
**Confidence:** 4

**Summary:**

This paper presents HawkBench, a new benchmark designed to evaluate the robustness of Retrieval-Augmented Generation (RAG) systems across varying levels of information-seeking tasks. The authors categorize queries into four levels, ranging from straightforward explicit fact-based questions to more complex implicit reasoning questions. To construct the benchmark, they sample diverse corpora from domain-specific areas and use GPT-4o or DeepSeek-v3 to generate initial QA pairs aligned with the four question levels. Human validation is conducted to ensure the quality and integrity of the data. Experimental results on both standard and enhanced RAG methods reveal that no single approach performs consistently well across all levels of HawkBench, highlighting the need for more generalizable and advanced RAG systems.

**Dataset Code Accessibility:**

Yes

**Dataset Code Comments:**

This paper has provided links to the dataset and code for reproduction.

**Ethical Comments:**

The sourced datasets are publicly accessible online.

**Ethical Considerations:**

No, there are no or only very minor ethics concerns

**Final Justification:**

Overall, the author has addressed most of my concerns, provided that the revisions are reflected in the manuscript. I am open to adjusting my scores accordingly. The rationale for the score adjustment is as follows:

1. The authors clarified how the proposed S-F1 metric is computed, noting that it involves the use of an LLM-as-a-judge. However, the paper does not mention this at all, and a revision is needed to explicitly acknowledge and explain this component.
2. The authors presented data contamination results for the legal, finance, and science domains, demonstrating that HawkBench retains robustness for evaluating RAG performance from a stratified perspective. However, contamination analyses for the remaining domains in HawkBench are not provided. Additionally, the benchmark lacks a baseline setting in which no retrieved documents are included, which limits the completeness of the evaluation.

**Limitations Weaknesses:**

1. I noticed that the source corpora are publicly accessible. Although the data are domain-specific, there is a possibility that some of the context was seen by the LLMs during pre-training. If that is the case, it becomes difficult to distinguish whether the model's performance is due to genuine retrieval or mere memorization. Additionally, the paper lacks a baseline setting where no retrieved documents are provided.
2. The paper does not provide a detailed explanation of how the S-F1 score is computed. The phrase “supported by the predicted answer” is vague, and the implementation details are unclear.
3. The proposed benchmark is not easily scalable, as it relies heavily on manual human annotation.
4. The generated questions and answers are relatively short, which limits the evaluation of long-form queries and the capabilities of RAG systems in handling complex reasoning.
5. The prompts used to generate the question-answer pairs are not provided.

**Strengths Contributions:**

1. A new high-quality QA benchmark for RAG is proposed, featuring questions of varying difficulty levels to evaluate the resilience of RAG systems across different domains and complexities.
2. Extensive experiments are conducted, covering various types of RAG methods, along with analyses of inference latency, the number of retrieved documents, and different retriever types.
3. The paper is well-written and easy to follow.

---

> ### Author Rebuttal · Authors · 2025-07-30
>
> Thank you for your thorough and constructive review. We greatly appreciate your recognition of HawkBench’s quality, coverage, and detailed experimental analysis. We address each of your comments and suggestions below:
>
> ---
>
> ### 1. Public Data and Model Memorization vs. Genuine Retrieval
>
> Thank you for highlighting this important concern. We fully acknowledge that most publicly available web data—including domain-specific corpora—are likely included in the pre-training corpus of today’s large language models. This challenge is shared by nearly all modern NLP benchmarks.
>
> However, we believe that benchmarks built on these corpora remain meaningful for several reasons. Seeing a text during pre-training does not guarantee full memorization, nor does it ensure accurate answers for queries requiring complex reasoning or synthesis. Our benchmark’s query and answer pairs are manually annotated to capture nuanced, multi-step information-seeking behaviors that go well beyond simple fact recall.
>
> To further mitigate concerns about model memorization versus genuine retrieval, we have included evaluations using several strong commercial LLM APIs. By comparing the performance of a range of models on our benchmark, we can better assess the degree to which retrieval-augmented reasoning (rather than memorization) contributes to success.
>
> Specifically, we compare RAG methods with three leading commercial LLM APIs. The results are summarized below.
>
> | Domain & Level | gemini-2.5-flash | gpt-4o-mini | gpt-4.1 | Best RAG (using Qwen-2.5 7B)      |
> | ----- | ---------------- | ----------- | -------------------- | -------- |
> | Legal-Level 1 | 13.6             | 10.5        | 15.3                 | **79.2** |
> | Legal-Level 2 | 25.2             | 20.6        | 30.0                    | **45.9** |
> | Legal-Level 3 | 15.8             | 20.7        | 22.0                  | **32.7** |
> | Legal-Level 4 | 12.4             | 13.2        | 14.3                   | **17.9** |
> | Finance-Level 1 | 13.6             | 12.3        | 22.3                   | **79.5** |
> | Finance-Level 2 | 25.2             | 13.3        | 23.1                    | **54.7** |
> | Finance-Level 3 | 13.8             | 17.8        | 20.2                    | **30.8** |
> | Finance-Level 4 | 13.4             | 15.8       | 18.9                    | **20.4** |
> | Science -Level 1 | 13.6             | 12.2        | 13.2                    | **45.1** |
> | Science -Level 2 | 25.1             | 17.8        | 21.2                     | **33.8** |
> | Science -Level 3 | 15.7             | 21.4        | 20.1                  | **27.3** |
> | Science -Level 4 | 13.4             | 17.2      | 17.4              | **20.4** |
>
> The results indicate that while strong LLMs have memorized substantial information from public corpora, they still lag behind retrieval-augmented methods in overall performance. Notably, for factoid queries, RAG methods outperform strong LLMs by a significant margin, suggesting that even with exposure to the underlying texts during training, LLMs cannot reliably recall fine-grained factual details. For Level-3 and Level-4 tasks, which require summarizing broad content or synthesizing information, strong LLMs perform comparably to RAG methods, as these queries demand less precise retrieval and more general reasoning.
>
> Overall, these results demonstrate that **even though portions of HawkBench may have been seen by strong LLMs during pretraining, it remains a robust benchmark for evaluating stratified RAG performance. Without retrieval, even advanced LLMs like GPT-4.1 can only solve a small fraction of the tasks, highlighting the necessity of effective retrieval-augmented reasoning.** Additionally, these experiments inspire us that HawkBench serves not only as a comprehensive testbed for RAG evaluation, but also as a tool for assessing the factual memorization capabilities of state-of-the-art LLMs.
>
> ---
>
> ### 2. Explanation of the S-F1 Score
>
> Thank you for raising this concern. We apologize for any lack of clarity in our description of the S-F1 metric, and are happy to provide additional details.
>
> For Level-3 and Level-4 tasks involving rationale queries, we introduce the **S-F1** metric to robustly assess sentence-level semantic equivalence between the ground-truth and predicted answers. Specifically:
>
> - For each sentence in the predicted answer, we use a strong LLM judge to check whether its meaning is supported by the ground-truth answer (precision).
> - For each sentence in the ground-truth answer, we similarly check whether it is supported by the predicted answer (recall).
> - S-F1 is computed as the average of these sentence-level precision and recall values.
>
> “Supported by the predicted answer” means that, for a given sentence in the ground-truth (or predicted) answer, the LLM judge determines if the meaning or rationale of that sentence is present—possibly rephrased but semantically equivalent—in the other answer.
>
> More specifically, we tokenize the predicted answer into individual sentences, and use the following prompt to determine a binary (0/1) score for each sentence. The same process is applied to each sentence in the ground-truth answer, with roles reversed.
>
> ```
> You are an evaluator assessing whether a sentence is semantically contained within a longer reference answer. Your task is to determine if the key meaning and information of the sentence is fully expressed somewhere within the reference answer, and that the sentence does not contain any additional information beyond what is in the reference.
>
> Follow these rules:
> 1. The sentence must be fully semantically contained - all key information and implications must be present in the reference
> 2. The sentence must not contain any information or implications that go beyond what is stated in the reference
> 3. The information can be spread across multiple sentences in the reference
> 4. Different wording or phrasing is acceptable as long as the core meaning is preserved
> 5. Provide only 1 / 0 as your response
>
> Example:
> Reference Answer: The Industrial Revolution began in Britain in the late 18th century. It led to major technological advances in manufacturing and transportation. This period saw rapid urbanization as people moved to cities for factory work.
>
> Sentence to check: People moved from rural areas to cities to work in factories during the Industrial Revolution.
> Response: 1
>
> Sentence to check: The Industrial Revolution caused urbanization and social upheaval.
> Response: 0 (Contains information about social upheaval not mentioned in reference)
>
> Reference Answer: {reference}
> Sentence to check: {sentence}
>
> Provide only 1 / 0:
> ```
>
> This prompt can be found in our code repository.
>
> We will revise the manuscript to clarify both the conceptual definition and practical computation of S-F1, as well as the role of the LLM judge in sentence-level evaluation.
>
> ---
>
> ### 3. Scalability and Human Annotation
>
> Thank you for your comment. We acknowledge that our benchmark construction relies on substantial human annotation, which is standard practice for ensuring the quality and reliability of high-quality evaluation datasets. Most leading NLP and IR benchmarks depend on expert or crowd-sourced annotation for the same reason.
>
> To address scalability, we have fully disclosed all annotation procedures and guidelines. We also provide our annotation system and supporting scripts, which greatly reduce the time and cost for future annotation efforts. This infrastructure makes it straightforward to scale HawkBench to larger datasets or new domains as needed.
>
> We will emphasize these points in the revised manuscript to highlight both our commitment to quality and the practical scalability of our approach.
>
> ---
>
> ### 4. Short Answers vs. Long-Form Evaluation
>
> Thank you for raising this point. We recognize that our current benchmark primarily features questions and answers of moderate length. This design choice was intentional: shorter answers are more easily verifiable and enable a more objective, high-quality evaluation—an approach also adopted by most widely-used benchmarks such as KILT [1], HLE [2], and GAIA [3].
>
> Nevertheless, our annotation pipeline fully supports the generation of long-form questions and answers if more complex, multi-step reasoning is desired. We opted for shorter forms to ensure answer verifiability and minimize reliance on subjective judgments, which are often required for evaluating longer, open-ended responses and typically need LLM-as-judge or extensive human review.
>
> We will clarify this rationale in the manuscript, and our released annotation tools can be used to create and validate long-form answers if the community wishes to extend the benchmark.
>
> **References:**
> [1] KILT: a Benchmark for Knowledge Intensive Language Tasks
> [2] Humanity’s Last Exam
> [3] GAIA: A Benchmark for General AI Assistants
>
> ---
>
> ### 5. Release of QA Generation Prompts
>
> We provide all built-in prompts used to generate the question-answer pairs, along with the complete source code, in our public repository. This is noted in lines 197–198 of the manuscript to ensure full transparency and reproducibility.

---

> > ### Comment · Reviewer_kbUp · 2025-08-05
> > **Official Comment by Reviewer kbUp**
> >
> > Thanks authors for their response.
> >
> > > 1.Public Data and Model Memorization vs. Genuine Retrieval
> >
> > I agree that data contamination is a common concern across modern benchmarks, which is why it is important to conduct thorough contamination analyses for newly proposed datasets. Such analyses help demonstrate that the benchmark remains minimally affected or still offers meaningful value to the research community.
> >
> > I appreciate the authors' efforts in showing that HawkBench maintains robustness for evaluating RAG performance from a stratified perspective, even though with partial data exposure during pre-training. That said, I would encourage the inclusion of complete contamination analysis results across all domains, not just legal, finance, and science, in the final revision. In addition, it would be good to have a baseline setting where no retrieved documents are provided.
> >
> > > 2.Explanation of the S-F1 Score
> >
> > Thank you for the clarification. I now have a better understanding of how the S-F1 score is computed.
> >
> > However, it remains unclear which judge model was used for the evaluation. The paper does not mention the use of an LLM-as-a-judge anywhere, which is why I raised questions about how the scoring was conducted. I recommend revising the paper to clarify this point.
> >
> > Overall, the author has addressed most of my concerns, provided that the revisions are reflected in the manuscript. I am open to adjusting my scores accordingly.

---

> > > ### Author Response · Authors · 2025-08-05
> > >
> > > Thank you very much for your thoughtful follow-up and for recognizing our clarifications during the rebuttal phase. We would make further clarifications below:
> > >
> > > - Public Data Contamination
> > >
> > > We appreciate your emphasis on the importance of contamination analysis. During the rebuttal phase, we conducted additional contamination analyses across all domains, not just legal, finance, and science. Due to the character limit, we were unable to provide the full set of results previously, but we will include these comprehensive analyses in our revised manuscript.
> > > Additionally, as you suggested, we will also add a non-retrieval baseline setting to better disentangle the effects of memorization from genuine retrieval-augmented reasoning.
> > >
> > > - Clarification of the S-F1 Judge Model
> > >
> > > Thank you for highlighting the need to clarify the LLM judge used for S-F1 evaluation. We indeed use DeepSeek-V3 as our LLM-as-a-judge model for all S-F1 scoring. This important detail was missing in the initial manuscript, and we will make sure to clearly state it in the revised version.
> > >
> > > Once again, thank you for your constructive feedback and openness to updating your evaluation. We are committed to incorporating all these improvements in the final manuscript.

---

> ### Author Response · Authors · 2025-08-04
>
> Dear Reviewer kbUp, we sincerely appreciate your thoughtful comments and suggestions. We have carefully addressed your concerns in our rebuttal. As the rebuttal period nears its end, we kindly ask you to review our responses and let us know if they have satisfactorily resolved your issues.

---

### Official Review · Reviewer_7HiR · 2025-07-03

**Rating:** 6
**Confidence:** 5

**Summary:**

The paper introduces a well-structured benchmark that addresses a clear gap in the current RAG evaluation landscape, with comprehensive task coverage, corpus consistency, and rigorous evaluation design from the aspects of difficulty levels and domains.
It provides comparative analyses across multiple types of RAG systems, revealing model-specific strengths and limitations across task types, thus offering high diagnostic value for the community.

**Dataset Code Accessibility:**

Yes

**Ethical Considerations:**

No, there are no or only very minor ethics concerns

**Final Justification:**

I think this is a well-qualified work and would interest the community. I will keep the high score.

**Limitations Weaknesses:**

1. The differences compared with existing RAG benchmark could be further emphasized, which can highlight the necessity of proposing current benchmark.
2. The selection of the evaluated RAG system could be explained more to align with the motivation of the proposed RAG benchmark, e.g., why enhanced, and global RAG and their differences.
3. The evaluation on retrieval part could be added, e.g., evaluating different retriever models and the impact of different search results for RAG, which is crucial in RAG tasks.

**Strengths Contributions:**

1. Unlike existing benchmarks that focus on limited task types, the proposed benchmark covers different types of queries which is categorized into four difficulty levels, showing better coverage for a range of information-seeking needs. Covering more query types is important in information-seeking benchmark. Besides, the benchmark incorporates diverse data sources (legal, financial, scientific, etc.), reducing domain bias and making it more representative of real-world retrieval challenges.
2. The authors introduce S-F1, a semantic-metric to provide a more nuanced assessment of reasoning-heavy queries. The evaluation protocols and annotation quality control enhance the data reliability.
3. Thorough evaluation on a set of RAG systems, including vanilla, enhanced, and global RAG methods, which have been conducted with associated detailed analysis by identifying their strengths and weaknesses across different task levels, making the benchmark more valuable. The results emphasize the importance of dynamic task adaptation, global reasoning strategies, and the potential of agentic RAG frameworks for adaptive information retrieval.

---

> ### Author Rebuttal · Authors · 2025-07-30
>
> Thank you for your detailed and thoughtful review. We greatly appreciate your recognition of HawkBench’s comprehensive task coverage, multi-domain design, rigorous evaluation, and diagnostic value for the RAG community. We address your specific comments and suggestions below:
>
> ---
>
> ### 1. Differences from Existing RAG Benchmarks
>
> Thank you for highlighting the need to further emphasize the distinction between HawkBench and existing RAG benchmarks.
>
> Most widely-used public RAG benchmarks focus on narrow or isolated aspects of information-seeking tasks. For example, LegalBench-RAG [1] evaluates legal queries, MultiHop-RAG [2] focuses on multi-hop reasoning, and CRAG [3] targets factual QA. While effective within their domains, these benchmarks collectively suffer from several key limitations:
>
> - **Fragmented evaluation protocols:** Existing benchmarks are typically siloed by query type, resulting in inconsistent criteria that hinder fair, comprehensive comparison of RAG performance across task categories.
> - **Domain bias and knowledge leakage:** Many use heterogeneous sources (e.g., Wikipedia, web snippets) that LLMs are likely pretrained on, leading to corpus-dependent gaps and inflated scores due to memorization rather than true retrieval-augmented reasoning.
> - **Limited query diversity:** Factoid questions dominate, while rationale-based queries—requiring synthesis and contextual analysis—are often neglected. This misaligns with real-world information-seeking, which spans both factual lookup and complex reasoning.
>
> HawkBench addresses these gaps by offering a unified, human-labeled, multi-domain benchmark designed to systematically evaluate RAG resilience across stratified information-seeking tasks. Our benchmark provides a consistent evaluation framework enabling fair and comprehensive comparison of diverse RAG methods.
>
> We will further highlight these distinctions and the necessity of HawkBench in the revised manuscript.
>
> ---
>
> ### 2. Rationale for Evaluated RAG Systems
>
> Thank you for suggesting clarification on the choice of evaluated RAG baselines.
>
> As shown in Table 4 (Appendix), we report results for a set of carefully selected baselines that represent the primary design paradigms in current RAG research:
>
> - **Vanilla RAG:** The standard pipeline, retrieving evidence and directly feeding it into the LLM as context (mainly varying the number of retrieved documents, i.e., top-k).
> - **Enhanced RAG:** These systems enrich query semantics via techniques like query rewriting, hypothesis generation, or other reformulations, with the goal of improving retrieval relevance and downstream reasoning.
> - **Global RAG:** Such methods convert the full knowledge base into structured forms (e.g., graphs, memory modules) prior to retrieval, enabling the model to reason over global knowledge structures rather than only local context.
>
> By covering these representative RAG paradigms, our evaluation enables systematic comparison of performance and adaptation across the diverse, stratified tasks in HawkBench. We will make these distinctions and motivations more explicit in the revised paper.
>
> ---
>
> ### 3. Evaluation of Retrieval Methods
>
> We fully agree that retrieval strategy is a crucial aspect of RAG system performance and include detailed analyses from two perspectives:
>
> - **Retrieval strategy:**
>   We evaluate dense retrieval, sparse retrieval, and a hybrid approach that combines both, across multiple representative RAG methods. Results (Figure 5) show that dense and hybrid retrievers consistently outperform sparse retrievers—especially on rationale-intensive tasks—while methods with additional retrieval cues (e.g., query rewriting, memory-guided retrieval) benefit particularly from hybrid strategies, which enhance coverage of relevant evidence.
>
> - **Number of retrieved documents (Top-k):**
>   Figure 4 presents the effect of varying Top-k. Increasing Top-k introduces more knowledge, but also more redundancy. The optimal recall-precision trade-off depends on query type:
>   - For factoid queries, too much redundancy degrades precision and performance.
>   - For rationale queries, higher recall (larger Top-k) is beneficial as synthesizing comprehensive answers requires broader evidence aggregation.
>
> We will further clarify these retrieval evaluations and their implications for RAG development and assessment in the revised manuscript.
>
> [1] LegalBench-RAG: A Benchmark for Retrieval-Augmented Generation in the Legal Domain
>
> [2] MultiHop-RAG: Benchmarking Retrieval-Augmented Generation for Multi-Hop Queries
>
> [3] Corrective Retrieval Augmented Generation

---

> > ### Comment · Reviewer_7HiR · 2025-08-05
> >
> > Look forward to seeing your revised version!

---

### Author Response · Authors · 2025-08-08
**General response**

Dear Reviewers, AC, SAC, and PC,

We would like to express our sincere gratitude for your careful review and constructive feedback. We are encouraged by the positive reception and thoughtful comments regarding our work.

We are pleased that reviewers recognized the key strengths of our paper, including:

- The design of a high-quality, human-annotated benchmark that systematically covers a broad range of information-seeking needs, spanning multiple domains and four levels of query difficulty, thereby offering a more comprehensive and realistic evaluation of RAG systems than existing datasets;
- The integration of diverse data sources—such as legal, financial, scientific, and literary texts—which reduces domain bias and better represents real-world retrieval challenges;
- The introduction of S-F1, a novel semantic metric that enables nuanced and reliable evaluation of reasoning-heavy queries;
- The robust experimental framework, including side-by-side comparisons of vanilla, enhanced, and global RAG systems, detailed ablation studies, and analysis of inference efficiency, all of which provide actionable insights for improving adaptive information retrieval;
- The paper’s clear structure, systematic annotation process, and transparent methodology, making it accessible and valuable to the broader research community.

We are especially encouraged that reviewers highlighted HawkBench’s practical value, broad coverage, and its ability to reveal the strengths and weaknesses of various RAG setups across different tasks and domains.

We are also grateful for the ethics reviewers’ thorough inspection of our paper’s ethical considerations. Both reviewers confirmed the ethical integrity of our work and its adherence to relevant regulations. We appreciate the actionable suggestions for further improvement. We will incorporate these clarifications and supplementary details into the final version to further strengthen our transparency and methodological rigor.

**We are greatly encouraged that reviewers 7HiR and kENp have expressed a highly positive attitude toward our paper from the beginning of the rebuttal phase.** During the rebuttal phase, we carefully addressed all reviewer concerns and questions. **We are glad to have received confirmation from all reviewers that their concerns have been resolved, and we appreciate the willingness of reviewers kbUp and xziF to update their initial assessments.** The additional analyses and clarifications provided in our rebuttal have further strengthened the paper.

We thank all reviewers for their invaluable feedback and guidance. In the final version, we will integrate the supplementary results, ethics-related clarifications, and methodological improvements discussed during the rebuttal.

---

### Decision · Program_Chairs · 2025-09-18

**Decision:**

Accept (spotlight)

**Comment:**

This paper presents HawkBench, a human-labeled, multi-domain benchmark designed to rigorously assess RAG performance across categorized task types. Reviewers all appreciated that the benchmark was of high quality with human annotations, its coverage of different types of queries of various difficulty levels, the design of the semantic metric, and the analysis of existing RAG systems. Rebuttal has clarified most concerns -- please incorporate them into the final paper.